# Intruding with Words: Towards Understanding Graph Injection Attacks at the Text Level

**Runlin Lei**
Renmin University of China
runlin_lei@ruc.edu.cn

**Yuwei Hu**
Renmin University of China
huyuweiyisui@ruc.edu.cn

**Yuchen Ren**
Renmin University of China
siriusren@ruc.edu.cn

**Zhewei Wei** *
Renmin University of China
zhewei@ruc.edu.cn

## Abstract

Graph Neural Networks (GNNs) excel across various applications but remain vulnerable to adversarial attacks, particularly Graph Injection Attacks (GIAs), which inject malicious nodes into the original graph and pose realistic threats. Text-attributed graphs (TAGs), where nodes are associated with textual features, are crucial due to their prevalence in real-world applications and are commonly used to evaluate these vulnerabilities. However, existing research only focuses on embedding-level GIAs, which inject node embeddings rather than actual textual content, limiting their applicability and simplifying detection. In this paper, we pioneer the exploration of GIAs at the text level, presenting three novel attack designs that inject textual content into the graph. Through theoretical and empirical analysis, we demonstrate that text interpretability, a factor previously overlooked at the embedding level, plays a crucial role in attack strength. Among the designs we investigate, the Word-frequency-based Text-level GIA (WTGIA) is particularly notable for its balance between performance and interpretability. Despite the success of WTGIA, we discover that defenders can easily enhance their defenses with customized text embedding methods or large language model (LLM)–based predictors. These insights underscore the necessity for further research into the potential and practical significance of text-level GIAs. The code is available at https://github.com/Leirunlin/Text-level-Graph-Attack.

## 1 Introduction

Graph Neural Networks (GNNs) have exhibited exceptional performance in various tasks [30, 34]. However, their vulnerability to graph adversarial attacks has been increasingly exposed. Research shows that Graph Modification Attacks (GMAs), which perturb a small portion of edges and node features of the original graph, can significantly degrade the performance of GNNs [37, 38]. Besides GMAs, Graph Injection Attacks (GIAs) generate harmful nodes and connect them to the original nodes, thereby reducing the performance of GNNs. Since adding new content to the public dataset is more practical than modifying existing content, GIAs are considered one of the most realistic graph adversarial attacks [28, 33].

---

*Zhewei Wei is the corresponding author. The work was partially done at Gaoling School of Artificial Intelligence, Beijing Key Laboratory of Big Data Management and Analysis Methods, MOE Key Lab of Data Engineering and Knowledge Engineering, and Pazhou Laboratory (Huangpu), Guangzhou, Guangdong 510555, China.

38th Conference on Neural Information Processing Systems (NeurIPS 2024).

Text-Attributed Graphs (TAGs), characterized by nodes with text-based features, form a crucial category among various graph types. In the early stages of GNNs, textual node feature processing is decoupled from the network design. Researchers transform text into fixed embeddings to investigate GNN designs on downstream tasks [13, 8, 27]. However, recent advancements have shifted toward a non-decoupled framework that integrates raw text into the design, significantly improving the ability to capture textual characteristics and enhance performance [9, 10]. Despite this progress in GNN design, current GIAs still focus on fixed embedding, aiming at only the injection of node embeddings rather than raw text on TAGs. This leads to several practical issues: **1) The attack setting is unrealistic.** In real-world scenarios like social or citation networks, it is more practical to inject nodes with raw text rather than node embeddings. For example, to attack a citation network dataset, it would be more natural to upload fake papers rather than a batch of embeddings. **2) The injected embeddings may be uninterpretable.** As minor perturbations at the embedding level can lead to unpredictable semantic shifts, traditional GIAs fail to ensure that injected nodes convey understandable semantic information. **3) Increased Detectability of Attacks.** Attackers usually have access only to raw text, not the processed embeddings defenders use. This discrepancy makes it challenging for attackers to mimic the embedding methods of defenders, leading to structurally abnormal embeddings. For example, embeddings based on word frequency are sparse, while those from Pre-trained Language Models (PLMs) are typically output continuous and dense. If an attacker injects dense embeddings while a defender uses a word-frequency-based model, the difference in the embedding structure makes the injected embeddings easy to detect.

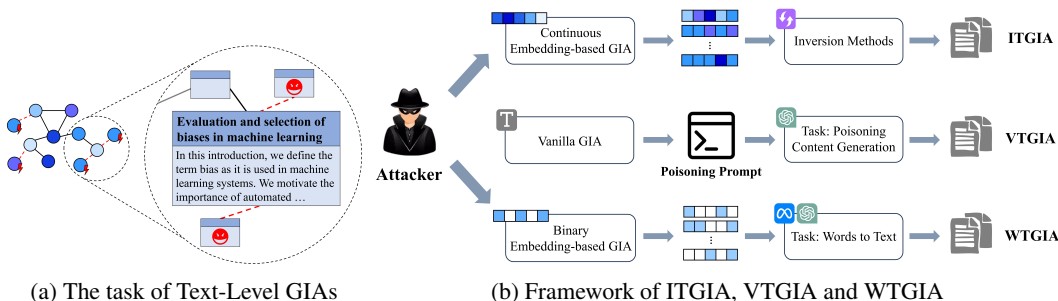

(a) The task of Text-Level GIAs      (b) Framework of ITGIA, VTGIA and WTGIA

Figure 1: Illustration of the Text-Level GIA setup and the three designs explored.

To address the above limitations, in this paper, we innovatively explore text-level GIAs, comprehensively examining their implementation, performance, and challenges. We explore three text-level GIAs: Vanilla Text-level GIA (VTGIA), Inversion-based Text-level GIA (ITGIA), and Word-frequency-based Text-level GIA (WTGIA), all of which successfully inject raw text into graphs and degrade the performance of GNNs, as shown in Figure 1b. Experiment results indicate that interpretability presents a significant trade-off against attack performance. ITGIA struggles to invert embeddings to interpretable text because the interpretable regions of the injected embeddings are ill-defined. VTGIA, which solely relies on Large Language Models (LLMs) for text generation, compromises attack effectiveness for its pursuit of text interpretability. WTGIA, as a balanced approach, manages to produce harmful and coherent text more effectively. However, the trade-off between performance and interpretability, coupled with poor transferability to different embedding methods, limits the performance of text-level GIAs in practice. Moreover, LLM-based predictors are shown to significantly enhance defenses against text-level GIAs, highlighting the substantial progress still needed to refine these attacks. In summary, our contributions are as follows:

- To the best of our knowledge, we are the first to unveil the dynamics of text-level graph adversarial attacks on TAGs, especially for GIAs. We discuss the impracticality of embedding-based GIA in real-world applications and propose a more realistic attack setting.

- We propose three effective text-level GIAs and demonstrate the trade-off between attack performance and text interpretability from both theoretical and empirical perspectives, providing insights for further refining text-level GIAs.

- We reflect on the challenges of graph adversarial attacks at the text level. We discover that simple strategies at the text level can enhance defense performance against text-level GIAs, which reemphasizes the importance of exploring GIAs at the text level.

## 2 Background and Preliminaries

A Text-Attributed Graph (TAG) is represented as $\mathcal{G} = (\mathcal{V}, \mathcal{E}, \{s_i\})$, where $\mathcal{V}$ is the node set and $\mathcal{E}$, the edge set. We denote $N$ as the number of nodes and $\mathbf{A} \in \{0, 1\}^{N \times N}$ as the adjacency matrix. For each node $v_i \in \mathcal{V}$, a sequential text feature $s_i$ is associated. We focus on semi-supervised node classification tasks, where each node is categorized into one of $C$ classes. The node labels are denoted by $\boldsymbol{y} \in \{0, \ldots, C-1\}^N$. The task is to predict test labels $\mathbf{Y}_{\text{test}}$ based on $\mathcal{G}$ and training labels $\mathbf{Y}_{\text{train}}$.

**Graph Neural Networks.** We use $f_\theta$ to denote a GNN parameterized by $\theta$. Generally, raw text $\{s_i\}$ is first embedded into a feature matrix $\mathbf{X} \in \mathbb{R}^{N \times F}$ using word embedding techniques, where $F$ is the output dimension. In the rest of our paper, when referring to vanilla GNNs that take text embeddings as inputs, we represent the text-attributed graph as $\mathcal{G} = (\mathcal{V}, \mathcal{E}, \mathbf{X})$ and express $f$ as $f(\mathbf{X}; \mathbf{A})$. The update process of the $l$-th layer of $f$ can be formally described as:

$$h_i^l = f^l \left( h_i^{l-1}, \text{AGGR} \left( \{h_j^{l-1} : j \in \mathcal{N}_i\} \right) \right), \tag{1}$$

where $h_i^0 = \mathbf{X}_i$, $\mathcal{N}_i$ is the neighborhood set of node $v_i$, AGGR is the aggregation function, and $f^l$ is a message-passing layer that takes the features of $v_i$ and its neighbors as inputs.

**Graph Injection Attacks and Related Works.** Graph injection attacks aim to create a perturbed graph $\mathcal{G}' = (\mathcal{V}', \mathcal{E}', \mathbf{X}')$ to disrupt the accuracy of a GNN on target nodes. Formally, denote the number of injected nodes as $N_{\text{inj}}$, and the number of nodes in the poisoned graph $\mathcal{G}'$ as $N'$. Then, the node set of $\mathcal{G}'$ can be formulated as $\mathcal{V}' = \mathcal{V} \cup \mathcal{V}_{\text{inj}}$, and:

$$\mathbf{X}' = \begin{bmatrix} \mathbf{X} \\ \mathbf{X}_{\text{inj}} \end{bmatrix}, \mathbf{A}' = \begin{bmatrix} \mathbf{A} & \mathbf{A}_{\text{inj}} \\ \mathbf{A}_{\text{inj}}^T & \mathbf{O} \end{bmatrix}, \tag{2}$$

where $\mathbf{X}_{\text{inj}} \in \mathbb{R}^{N_{\text{inj}} \times F}$, $\mathbf{A}_{\text{inj}} \in \{0, 1\}^{N \times N_{\text{inj}}}$, and $\mathbf{O} \in \{0, 1\}^{N_{\text{inj}} \times N_{\text{inj}}}$.

For the design of injected node features, one line of research focuses on statistical features to ensure unnoticeability, sacrificing attack strength by excluding learnable processes for features [25, 4]. In parallel, other works target learnable injected embeddings. Wang et al. [29] pioneer injecting fake nodes with binary features to degrade GNN performance. AFGSM [28] introduces an approximation strategy to linearize the surrogate model, efficiently solving the objective function.

Starting from KDDCUP 2020, research shifts towards injecting continuous node embeddings under the constraints in Equation (3) [33, 36, 2]. Denote the budget of injected nodes as $\Delta$, the maximum and minimum elements in the original feature matrix $\mathbf{X}$ as $x_{\max}$ and $x_{\min}$, and the maximum degree allowed for each injected node as $b$. The objective of these GIAs can be formulated as:

$$\min_{\text{G}'} \frac{|\{f(\text{G}')_i = \boldsymbol{y}_i, v_i \in \mathcal{V}_T\}|}{|\mathcal{V}_T|}$$
$$\text{s.t. } N_{\text{inj}} \leq \Delta, d_{\text{inj}} \leq b, \quad x_{\min} \cdot \mathbf{1} \leq_{\text{e}} \mathbf{X}_{\text{inj}} \leq_{\text{e}} x_{\max} \cdot \mathbf{1}, \tag{3}$$

where $\mathcal{V}_T$ is the target node set, $d_u$ is the degree for node $u$, and $\mathbf{X}_1 \leq_{\text{e}} \mathbf{X}_2$ indicates each element of $\mathbf{X}_1$ is less than or equal to the corresponding element in $\mathbf{X}_2$. Among continuous strategies, TDGIA [36] generates node features by optimizing a feature smoothness objective function. Chen et al. [2] enhance homophily unnoticeability by adding a regularization term to the objective function. These approaches show better performance while maintaining unnoticeability at the embedding level.

It is worth noting that the evaluation datasets for the aforementioned GIAs are **primarily TAGs** [33], e.g., such as Cora and CiteSeer. However, all the studies remain at the embedding level. In our experiments, we adhere to the structural budget detailed in Equation (3), omitting embedding-level constraints as we pioneer GIAs at the text level. More related works are provided in Appendix D.

**Evaluation Protocol.** In this paper, we explore GIAs in an **inductive**, **evasion** setting, following the framework established by [33] to ensure closeness to practical scenarios. Attackers have access to the entire graph and associated labels, except for the ground-truth labels of the test set and the victim models of the defenders. l Defenders train their models on clean, unperturbed training data and then use trained models to classify unseen test data.

**General Experiments Set-up.** We conduct experiments on three TAGs, Cora [17], CiteSeer [6] and PubMed [23], which are commonly used in evaluating GIAs. The dataset statistics and attack budgets are provided in Table 5 and Table 7 in the Appendix. We adopt full splits in [33] for each

dataset, which uses 60% nodes for training, 10% nodes for validation, and 30% for testing. For word embeddings techniques, we include **Bag-of-Words (BoW)** and **GTR** [19], a T5-based pre-trained transformer that generates continuous embeddings of unit norm. The default BoW method constructs a vocabulary using raw text from the original dataset based on word frequency, with common stop-words excluded. The embedding dimension of BoW is set to 500. In the experiments, we use a 3-layer vanilla GCN or a 3-layer homophily-based defender EGNNGuard following [2]. EGNNGuard defends homophily-noticeable attacks by cutting off message-passing between nodes below the specified similarity threshold. The hyperparameters and training details are provided in Appendix H.1. We repeat each experiment 10 times and report the average performance and the standard deviation. Unless otherwise specified, we **bold** the best (the lowest) results.

## 3 Text-Level GIAs: Interpretability Matters

Although GIAs are considered realistic graph attacks, traditional methods focus mainly on the embedding level, which can be overly idealized. Typically, attackers only have access to public raw data, such as paper content and citation networks, on platforms like arXiv, and defenders process this data more personally. Since attackers struggle to access embeddings trained by defenders, injecting nodes with suitable embeddings is challenging. Conversely, injecting new raw data into the public dataset is more realistic for attackers. This leads us to an important question: *Can we generate coherent text for injection that ensures the effectiveness of attacks while maintaining interpretability?*

### 3.1 Inversion-based Text-Level GIAs: Effective yet Uninterpretable

Embedding-level GIAs have demonstrated strong attack performance against GNNs [36, 2]. To achieve text-level attacks, a straightforward idea is to use inversion methods to convert embeddings back into text. Recent advancements in text inversion methods, such as Vec2Text [18], have enabled text recovery from PLM embeddings with a 66% success rate for examples averaging 16 tokens. This breakthrough opens new possibilities for achieving text-level GIA that is comparable in performance to those at the embedding level, which we term **Inversion-based Text-level GIA (ITGIA)**.

**Implementation of ITGIA.** ITGIA consists of two steps: generating poisoning and invertible embeddings, and then converting them into text. Since the effectiveness of current inversion methods has only been validated on normal text embeddings, it is crucial to ensure that generated embeddings lie within an interpretable and feasible region. However, defining these interpretable regions precisely is challenging. The constraint specified in Equation (3) limits generated embeddings to a cubic space, which may be overly broad for feasible embeddings. For example, PLMs typically produce embeddings located on a spherical surface with an L2-norm of 1, representing a significantly smaller region. To meet this fundamental requirement, we employ Projected Gradient Descent (PGD)[16], projecting each injected embedding onto the unit sphere after every feature update. To further address the challenge of defining interpretable regions, we incorporate Harmonious Adversarial Objective (HAO) [2] into the GIA objective function. Optimizing over HAO increases the similarity between injected embeddings and those of normal text from original datasets, thereby bringing the injected embeddings closer to interpretable regions. Details of ITGIA and HAO are provided in Appendix I.1 and H.1. Once the embeddings are obtained, we utilize the GTR inversion model in [18] with a 20-step correction process to revert embeddings back into text.

**Analysis of Performance and Interpretability.** Following [2], we use sequential injection frameworks, including the sequential variants of SeqGIA (Random injection), TDGIA, ATDGIA, MetaGIA, and AGIA as the embedding-level backbones. The results of ITGIA are displayed in Table 1. We can see the embeddings after inversion deviate significantly from the injected embeddings, as indicated by the low cosine similarity. This suggests substantial information loss during inversion, leading to poor attack performance. Figure 6 illustrates the underlying reason: although injected embeddings meet basic norm constraints, they struggle to fall within the much smaller interpretable embedding region, hindering accurate inversion. Although incorporating HAO enhances the inversion accuracy and improves attack performance, the improvement comes at the expense of embedding-level attack performance [2]. As a result, increasing HAO weights does not consistently yield better results, as shown in Figure 2. Furthermore, the generated text remains incoherent with high perplexity, as demonstrated in Appendix J. This renders real-world applications impractical and highlights the challenges faced by continuous embedding-level GIAs.

Table 1: Performance of GCN on graphs under ITGIA. Raw text is embedded by GTR before being fed to GCN for evaluation. "Avg. cos" represents the average cosine similarity between the embeddings of the inverted text and their corresponding original embeddings across five ITGIAs. "Best Emb." represents the best attack performance across the five variants at the embedding level.

| Dataset | Clean | HAO | Avg. cos | SeqGIA | MetaGIA | TDGIA | ATDGIA | AGIA | Best Emb. |
|---------|-------|-----|----------|--------|---------|-------|--------|------|-----------|
| Cora | 87.19 ± 0.62 | x | 0.14 | 74.16 ± 1.76 | **71.35 ± 1.14** | 76.52 ± 1.45 | 76.73 ± 1.46 | 72.25 ± 1.32 | 31.14 ± 0.05 |
| | | ✓ | 0.67 | 65.67 ± 1.48 | 65.77 ± 1.15 | 71.49 ± 1.71 | 74.63 ± 2.48 | 68.59 ± 1.51 | |
| CiteSeer | 75.93 ± 0.41 | x | 0.11 | 68.17 ± 0.94 | 69.39 ± 0.89 | 68.24 ± 1.30 | 69.72 ± 1.34 | **66.18 ± 1.19** | 21.45 ± 0.58 |
| | | ✓ | 0.56 | 64.79 ± 1.30 | 65.11 ± 1.01 | 67.43 ± 0.89 | 71.89 ± 0.50 | **64.79 ± 1.30** | |
| PubMed | 87.91 ± 0.26 | x | 0.06 | 65.13 ± 1.67 | **58.96 ± 1.25** | 59.49 ± 1.08 | 69.81 ± 1.90 | 66.16 ± 0.97 | 38.32 ± 0.00 |
| | | ✓ | 0.59 | 66.40 ± 2.33 | **58.56 ± 1.22** | 60.26 ± 1.32 | 76.23 ± 2.08 | 65.77 ± 0.91 | |

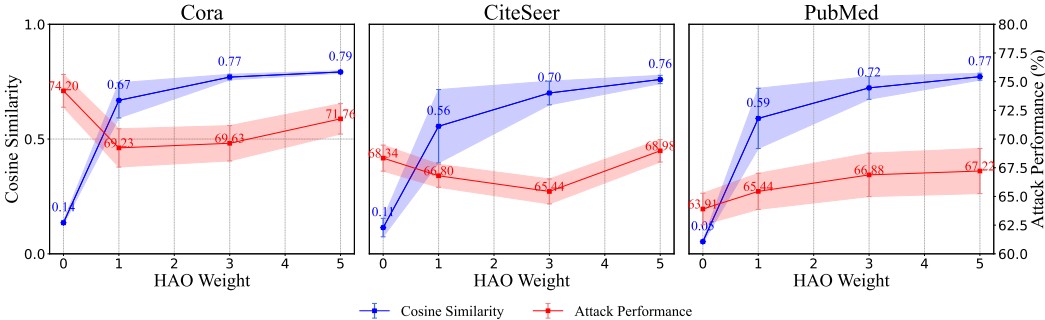

Figure 2: The change of attack performance as the weight of HAO increases. Lower performance stands for better attack results. Details of the setting is given in Appendix H.1.

## 3.2 LLM-based Text-Level GIAs: Interpretable yet Ineffective

Although ITGIA manages to generate harmful text, interpretability remains challenging to address. To ensure that the generated text is both interpretable and deceptive, a direct approach is to utilize an LLM for text generation and guide it to produce poisoning content through carefully designed prompts. Based on experiences with embedding-level graph adversarial attacks, when the features of injected nodes exhibit heterophily, confusion, or irrelevance, the classification performance of their neighboring nodes is negatively affected [35]. Thus, we attempt to incorporate prior knowledge used in traditional GIAs and create prompts from three different angles:

**Heterophily Prompt.** This prompt involves sampling the text of nodes from the original dataset to generate heterophilic (dissimilar) content. Specifically, we first conduct embedding-level GIAs to obtain a perturbed graph. Next, we sample the content from the neighbors of injected nodes. Finally, we design a prompt that requests the generated content to be dissimilar to their neighboring nodes, exploiting the weakness of GNNs in handling heterophily.

**Random Prompt.** This prompt generates node text that is entirely unrelated to the original categories of the graph. For example, in the Cora dataset, the original categories are about machine learning topics. In this prompt, we generate papers in categories like sports, art, and history, which are dissimilar to all nodes in the original graph.

**Mixing Prompt.** This prompt generates node text that may be classified into multiple or even all categories. Using Cora as an example, a potential paper title could be: "Unified Approach for Solving Complex Problems using Integrated Rule Learning, Neural Networks, and Probabilistic Methods".

After defining the prompts, we propose a simple Vanilla Text-level GIA (VTGIA). This method first generates text for the injected nodes based on the prompt via LLMs and subsequently optimizes the graph structure. We use GPT-3.5-1106 [20] as the LLM backbone for text generation, and the five sequential injection backbones same as ITGIA. The pseudo-code of VTGIA, prompts, and generated examples are provided in Appendix I.2, H.2, and J. The results are shown in Table 2. Although the generated content exhibits adversarial effects, directly producing harmful text via prompts is not sufficiently effective and is far from the optimal results achievable by attacks at the embedding level.

Table 2: Performance of GCN against VTGIA. Raw text is embedded by GTR before being fed to GCN for evaluation. "Best Emb." refers to the best-performing embedding-level GIAs that directly update embeddings across various injection strategies.

| Dataset | Clean | Prompt | SeqGIA | MetaGIA | TDGIA | ATDGIA | AGIA | Best Emb. |
|---------|-------|--------|--------|---------|-------|--------|------|-----------|
| Cora | 87.19 ± 0.62 | Heterophily | 83.35 ± 0.49 | **80.81 ± 0.37** | 84.23 ± 0.80 | 82.05 ± 0.88 | 83.88 ± 0.83 | 31.14 ± 0.05 |
| | | Random | 84.65 ± 1.11 | **82.32 ± 0.66** | 85.51 ± 0.81 | 84.73 ± 0.82 | 86.21 ± 0.77 | |
| | | Mixing | 83.10 ± 0.80 | **80.78 ± 0.66** | 83.89 ± 1.32 | 83.91 ± 1.73 | 84.19 ± 1.21 | |
| CiteSeer | 75.93 ± 0.41 | Heterophily | 74.91 ± 0.55 | **73.32 ± 0.39** | 75.50 ± 0.44 | 73.73 ± 1.04 | 74.62 ± 0.86 | 21.45 ± 0.58 |
| | | Random | 73.84 ± 0.79 | 73.28 ± 0.69 | 72.61 ± 1.30 | 71.43 ± 0.96 | **70.81 ± 1.27** | |
| | | Mixing | 75.29 ± 0.67 | **74.16 ± 0.51** | 74.61 ± 0.71 | 74.74 ± 1.15 | 74.87 ± 1.03 | |
| PubMed | 87.91 ± 0.26 | Heterophily | 80.80 ± 0.83 | 77.50 ± 0.52 | **75.41 ± 1.22** | 75.78 ± 0.77 | 82.36 ± 0.53 | 38.32 ± 0.00 |
| | | Random | 81.99 ± 2.34 | **78.34 ± 2.08** | 80.39 ± 2.87 | 82.26 ± 4.46 | 86.23 ± 0.87 | |
| | | Mixing | 81.27 ± 1.91 | **78.48 ± 1.59** | 78.62 ± 2.78 | 80.37 ± 1.99 | 85.44 ± 0.80 | |

## 4 Word-Frequency-based Text-Level GIAs

Based on previous analysis, to achieve an interpretable and effective GIA, the following conditions should be met: 1) Guidance from Embedding-level GIA to ensure effectiveness. 2) Utilization of LLMs to guarantee interpretability. 3) A well-defined LLM task formulation to minimize information loss during the embedding-text-embedding conversion process.

We address these conditions by employing word-frequency-based embeddings, specifically binary Bag-of-Words (BoW) embeddings, which offer several advantages: 1) Clear physical meaning. In binary BoW embeddings, a '1' indicates the presence of a word, while a '0' indicates its absence. 2) Clear task formulation. The task involves generating text containing specified words while excluding prohibited words, which can be effectively done via LLMs. 3) Controllable embedding-text-embedding process. As long as the text includes specified words and excludes prohibited words, the embeddings can be exactly inverted. Building on these advantages, we design the **Word-frequency-based Text-level GIA (WTGIA)**, leveraging BoW embeddings and the generative capabilities of LLMs. Its implementation consists of three steps: 1) Obtain binary injected embeddings. 2) Formulate embedding to text tasks for LLMs. 3) Perform multi-round corrections for LLMs.

### 4.1 Row-wise Constrained FGSM: Unnoticeable and Effective at the Embedding Level

**Implementaion of Row-wise Constrainted FGSM.** To generate binary embeddings, we adopt a row-wise constrained Fast Gradient Sign Method (FGSM) algorithm based on [28]. Specifically, we first obtain vocabulary based on BoW on the original dataset. Next, we set the injected embedding $\mathbf{X}_{\text{inj}}$ to all zeros. We then set a sparsity constraint $S$ on injected embeddings, that each embedding can retain at most $F' = \lfloor S * F \rfloor$ non-zero entries, where $F$ is the dimension of features. In each epoch, we define the *flippable set* as the entries in $\mathbf{X}_{\text{inj}}$ that satisfy the sparsity constraint. Based on the element-wise gradient of the loss concerning $\mathbf{X}_{\text{inj}}$, we flip the most significant $B$ entries from the flippable set, where $B$ is the batch size. The process stops when all rows run out of word budgets. [2]

Note that the sparsity budget $S$ is a hyper-parameter that limits the use of at most $F'$ words from the predefined vocabulary in a single text. Intuitively, a higher number of used words intensifies the embedding-level attack but can compromise text-level interpretability, as it becomes harder to integrate more specified words naturally in the generated text. Based on the row-wise constraints, we can formally establish the relationship between embedding-level performance and unnoticeability and text-level interpretability.

**Definition 1** (Single node GIAs towards one-hot embedding). *For a target node $v_t$ with embedding*

$x_t = [\overbrace{1, \cdots, 1}^{k}, 0, \cdots, 0] \in \{0, 1\}^F$, $k \ll F$, *the embedding indicates that the node uses $k$ words from a predefined vocabulary, while $F - k$ words are not used. We define the set containing the specified $k$ words as set $W_u$ and the set containing the $F - k$ prohibited words as set $W_n$. We assume words in $W_u$ are related to their corresponding class, whereas words in $W_n$ are associated with other classes. To attack node $v_t$, an adversarial node $v_i$ with embedding $x_i \in \{0, 1\}^F$ is injected.*

---

[2] We do not consider the co-occurrence constraint in [28] because we find the co-occurrence matrix is dense and ineffective in practice (see Appendix G.3).

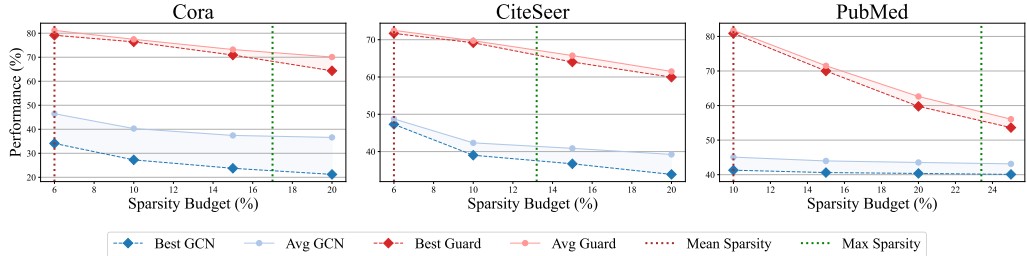

Figure 3: The Best and Average performance of FGSM against GCN and EGNNGuard among the five injection methods w.r.t increasing sparsity budgets. The results for EGNNGuard reveal that as the budget increases, FGSM attacks can satisfy the similarity constraint while significantly enhancing the attack performance at the embedding level.

- **Performance:** *An attack is more effective against node $v_t$ if it uses more words $w_n \in W_n$. The intuition is that mixing words from other classes makes $x_t$ harder to be correctly classified.*

- **Unnoticeability:** *An attack is less detectable if the similarity between $x_t$ and $x_i$ is higher.*

- **Interpretability:** *An attack is more interpretable if fewer words are required to use in $x_i$, which means $x_i \cdot \mathbf{1}$ is smaller. The intuition is that generating content with fewer specified words is easier, which fits in practical scenarios involving limited length.*

**Theorem 1.** *In the setting outlined in Definition 1, assume we apply a cosine similarity constraint with a threshold $c \in (0, 1)$ for unnoticeability. Specifically, this constraint requires that the cosine similarity between $x_t$ and $x_i$ satisfies $\frac{x_t \cdot x_i}{\|x_t\|\|x_i\|} > c$. Let $a$ denotes the number of words used by $x_i$ from the set $W_u$, and $b$ denotes the number of words used by $x_i$ from $W_n$. If the budget is $m$ words at most to ensure interpretability, then the maximum value of $b$ is $\max(b) = \max\left(\lfloor(m - c\sqrt{mk}\rfloor, 0\right)$.*

The proof and detailed illustration for intuitions are provided in Appendix F. Theorem 1 reveals that as long as $m \geq k$, we can find a positive value for $b$ (disregarding the flooring operation for simplicity), effectively introducing harmful information to the target node. Practically, given the typically low value of $c$, a destructive $x_i$ can be easily identified with a relatively large $m$ at the embedding level. In Figure 3, we present the average and best performance of Row-wise Constrained FGSM coupled with five sequential injection methods used in ITGIA and VTGIA against GCN and EGNNGuard. As the sparsity budget increases, the performance of EGNNGuard declines sharply, confirming that enhancements in performance are possible under the unnoticeability constraint. However, achieving significant performance degradation requires sacrificing interpretability. For example, to achieve a 10% performance drop in Cora and CiteSeer, FGSM attacks need a budget close to or exceed the maximum sparsity level of all texts belonging to the original datasets. The subsequent subsection will explore the consequences of sacrificing text-level interpretability while completing WTGIA.

## 4.2 Trade-off for Interpretability at the Text Level

**Task Formulation of LLMs.** After obtaining the binary injected embeddings, we can obtain the *Specified words* included and the *Prohibited words* excluded in each text based on the BoW vocabulary. Subsequently, we formulate the task for LLMs as a "word-to-text" generation problem. We request LLMs to generate text under a specified length limit using specified words, optionally within a given topic based on the origin dataset. [3] Handling prohibited words is more complex. Given the sparsity of BoW embeddings, the number of prohibited words is relatively large. Directly requesting the absence of prohibited words through the prompt would be challenging for LLMs to understand. So, for closed-source LLMs like GPT, we only constrain the specified words in the prompt. For open-source LLMs, we mask their corresponding entries during the output process.

**Masking Prohibited Words.** Denote $t_{i+1}$ as the next token at time $i$. The standard generative process for LLM can be formulated as: $t_{i+1} = \operatorname{argmax}\left(\operatorname{SoftMax}\left(\operatorname{logits}(t_{i+1} \mid t_i, t_{i-1}, \dots)\right)\right)$.

---

[3]The length requirement is necessary because the task would become trivial if the text length is unlimited. However, we usually cannot arbitrarily inject long text into real-world applications.

where logits$(t_{i+1} \mid t_i, t_{i-1}, \dots)$ are the unnormalized log probabilities of the potential next token given the previous tokens. To enforce constraints on avoiding prohibited words, we introduce a masking vector $M$ such that: $M[j] = 0$ if token $j$ is prohibited, and $1$ otherwise. The generative process is then modified by applying this mask to the logits before the SoftMax operation: $t_{i+1} = \text{argmax}\left(\text{SoftMax}\left(\text{logits}\left(t_{i+1} \mid t_i, t_{i-1}, \dots\right) \odot M\right)\right)$, where $\odot$ represents the element-wise multiplication of the logits by the mask. This mask effectively removes words from outputs by making their probabilities negligible.

**Multi-Round Correction.** After obtaining the output from the LLM, we calculate the use rate $\left(\frac{\text{Number of Specified Words used}}{\text{Number of Specified words}}\right)$ in the generated text and correct any omissions in their usage. Through multiple rounds of dialogue, we select the text with the highest use rate as the final output for LLMs.

**Main Results.** We utilized GPT3.5-turbo-1106 [20] as the close-source LLM and Llama3-8b [26] as the open-source LLM to implement the WTGIA. Based on Row-wise FGSM in Figure 3, we combine WTGIA with the five injection strategies and report the average results. The specific prompt details are in Appendix H.2. The length limit is specified in Table 6. The variants that use prompts that specify the generated content must belong to the original class set are marked as "-T." The results are displayed in Figure 4. It can be seen that

- **Desirable Performance:** WTGIA achieves comparable attack performance to Row-wise FGSM, demonstrating strong embedding-to-text capabilities.

- **Trade-off:** Variants with topic specification perform worse than those without, sacrificing attack strength for better text-level unnoticeability.

- **Masking Helps:** Llama-WM and GPT that do not mask prohibited words generally perform worse than Llama.

Complete experiments and full results are given in Appendix K and Appendix L. Overall, WTGIA effectively replicates the performance of FGSM at the embedding level, achieving both effectiveness and interoperability.

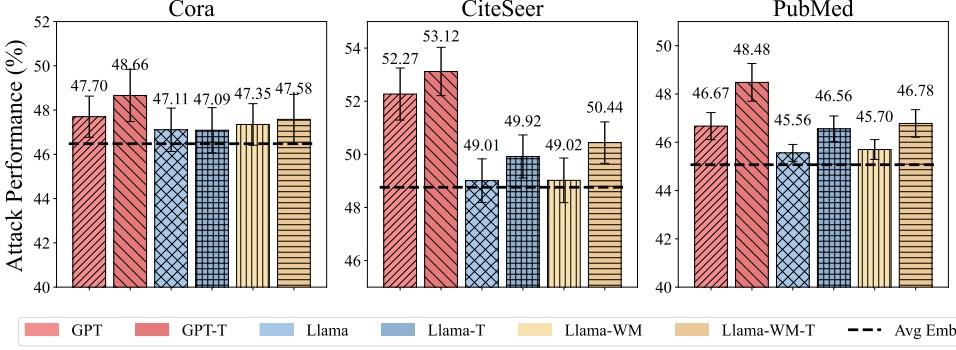

Figure 4: Performance of WTGIA against GCN. Sparsity budget is the average sparsity of the original dataset. Methods with -T include topic requirements in the prompt. Methods with -WM exclude masks for prohibited words in Llama. Avg Emb. represents the average FGSM attack performance at the embedding level. Lower values indicate better attack performance.

**New Trade-offs.** However, we fail to further enhance performance at the text level by increasing the sparsity budget, similar to FGSM at the embedding level. As shown in Figure 5, with the increasing sparsity budget, the use rate keeps decreasing, representing a trade-off to maintain the interpretability of the generated content. In Cora and CiteSeer, the text-level attacks perform best when sparsity is around 10%. In PubMed, GPT-Topic even fails to generate meaningful text with a sparsity budget $S \geq 15\%$. These observations confirm that **in text-level attacks, interpretability represents an additional trade-off that is overlooked in embedding-level attacks**. Enhancing or evaluating performance solely at the embedding level fails to provide a practical understanding of GIAs.

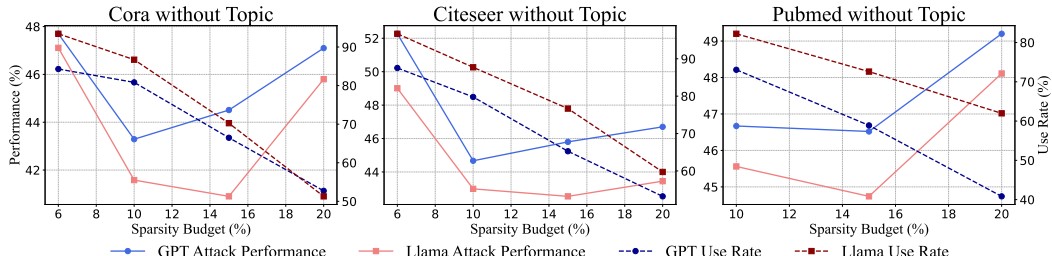

Figure 5: The performance of WTGIA w/o topic against GCN w.r.t sparsity budget. As the budget increases, the use rate keeps decreasing, and the attack performance increases and then decreases.

## 5 New Challenges for Text-level GIAs

### 5.1 Transferbility to Different Embeddings

In practical scenarios, defenders can employ any text embedding techniques they want to process the text once an attacker completes the injection. However, embedding-level attacks implicitly assume that defenders will use the same embeddings as the attackers during evaluation, potentially benefiting the attackers. In fact, it is crucial to fully consider the transferability of a text attack across different text embedding technologies for text-level attacks.

In Table 3, we present the results of ITGIA and WTGIA when transferred to different embeddings on the Cora dataset. When transferred to GTR embeddings, WTGIA does not exhibit a significant performance advantage over ITGIA. The performance of ITGIA when transferred to BoW embeddings is even poorer. Since the injected text is highly uninterpretable and deviates significantly from the original dataset, the BoW method inherently filters out uncommon terms, rendering the injected embeddings particularly sparse and ineffective. These limitations in embedding transferability underscore the research gaps between embedding- and text-level GIAs.

Table 3: Performance of ITGIA and WTGIA-Llama transferred to different embeddings on Cora.

| Text-GIA | Embedding | Clean | SeqGIA | MetaGIA | TDGIA | ATDGIA | AGIA |
|---|---|---|---|---|---|---|---|
| ITGIA | BoW | 86.48 ± 0.41 | 84.85 ± 0.76 | **84.04 ± 0.78** | 85.56 ± 0.61 | 86.49 ± 0.50 | 84.90 ± 0.73 |
| | GTR | 87.19 ± 0.62 | 66.70 ± 0.94 | 67.83 ± 0.75 | 71.49 ± 1.71 | 74.63 ± 2.48 | **68.81 ± 1.39** |
| WTGIA | BoW | 86.48 ± 0.41 | 48.32 ± 0.74 | 51.58 ± 0.78 | 52.49 ± 1.32 | **35.33 ± 1.29** | 47.81 ± 0.78 |
| | GTR | 87.19 ± 0.62 | 78.15 ± 1.70 | **76.88 ± 0.96** | 79.27 ± 1.24 | 83.77 ± 1.11 | 77.95 ± 1.51 |

### 5.2 LLMs as Defender

Since attacks are text-level, the defenders can directly utilize LLMs as predictors for node classification tasks without necessarily employing GNNs. Taking the LLMs-as-Predictor from [3] as an example. The method feeds the formulation of the node classification task, texts of target nodes, and optionally, the sampled texts of their neighbors in the prompt. It then utilizes LLMs to predict the labels of target nodes directly. Remarkably, this method demonstrates superior performance on clean datasets. We now examine the performance of WTGIA against LLMs-as-predictor from [3]. For WTGIA, we select the strongest variant in Figure 4 for each dataset as the attacker. In the zero-shot setting, only the text of target nodes is fed to LLMs. In the few-shot setting, the text of labeled nodes with their labels is fed to LLMs as examples. On Cora and CiteSeer, we adopt the whole test set for evaluation. For PubMed, we sample 1000 nodes from the test set for evaluation.

The results are presented in Table 4. Remarkably, on the PubMed dataset, the baseline without the use of neighborhood information ("Clean (w/o Nei)") achieves outstanding performance, which is consistent with observations in [3]. Therefore, even without utilizing any neighborhood information, LLM-based predictors can achieve high accuracy in node classification, meaning defenders can evade the influence of injected nodes. For other datasets, Cora and CiteSeer, while incorporating neighborhood information enhances the performance of LLMs-as-Predictors, the "Clean (w/o Nei.)"

Table 4: The performance of WTGIA against LLMs-as-predictor. The term "(w/o Nei.)" means the exclusion of neighborhood information in the prompt. Methods "Clean (w/o Nei.)" and "WTGIA (w Nei.)" can be used as LLM-based defenders. The best results for defenders are **bold**.

| Dataset | Zero-shot | | | Few-shot | | |
|---|---|---|---|---|---|---|
| | Clean (w Nei.) | Clean (w/o Nei.) | WTGIA (w Nei.) | Clean (w Nei.) | Clean (w/o Nei.) | WTGIA (w Nei.) |
| Cora | 78.64 | 67.90 | **74.81** | 79.51 | 66.54 | 72.71 |
| CiteSeer | 69.18 | 59.53 | 67.71 | 73.90 | 66.67 | **68.44** |
| PubMed | 89.80 | **89.80** | 89.30 | 84.50 | 80.00 | 80.20 |

baseline still represents a practical upper limit for the GIAs. Even if attackers manage to degrade the performance below this baseline, defenders can turn to this neighbor-free method as a secure fallback. Consequently, the flexibility of defenders should be fully considered when evaluating GIAs at the text level in practical scenarios.

## 6 Conclusion

In this paper, we explore the design of GIAs at the text level. We address the limitations of embedding-level GIAs on TAGs by extending them to the text level, which better aligns with real-world scenarios. We present three types of text-level GIA designs: ITGIA, VTGIA, and WTGIA. Our theoretical and empirical analysis reveals a trade-off between attack performance and the interpretability of the injected text. Among these methods, WTGIA achieves the best balance between performance and interpretability. Additionally, our findings indicate that defenders can effectively counter these attacks using different text embedding techniques or LLM-based predictors, highlighting the complex and challenging nature of graph adversarial attacks in practical applications.

## 7 Acknowledgement

This research was supported in part by National Natural Science Foundation of China (No. U2241212, No. 61932001), by National Science and Technology Major Project (2022ZD0114802), by Beijing Natural Science Foundation (No. 4222028), by Beijing Outstanding Young Scientist Program No.BJJWZYJH012019100020098, By Huawei-Renmin University joint program on Information Retrieval. We also wish to acknowledge the support provided by the fund for building world-class universities (disciplines) of Renmin University of China, by Engineering Research Center of Next-Generation Intelligent Search and Recommendation, Ministry of Education, Intelligent Social Governance Interdisciplinary Platform, Major Innovation & Planning Interdisciplinary Platform for the "Double-First Class" Initiative, Public Policy and Decision-making Research Lab, and Public Computing Cloud, Renmin University of China.

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

# A  Broader Impacts

In this paper, we potentially aid in developing more advanced and sophisticated attacks through understanding text-level vulnerabilities and providing significant positive impacts by enhancing the design of robust defense mechanisms. By pioneering the exploration of text-level GIAs, we enable the creation of more effective countermeasures to protect against these advanced threats, thereby improving the overall security and resilience of systems that rely on GNNs. This dual impact highlights the importance of our work in driving both offensive and defensive advancements, ultimately contributing to safer and more secure AI applications.

# B  Open Access to Data and Code

GPT-turbo: `https://openai.com/`. In this paper, we use the GPT-3.5-turbo model provided by OpenAI for generating textual data. The model was accessed through OpenAI's API services, and all uses were in compliance with the terms of service provided by OpenAI.

Llama3: `https://github.com/meta-llama/llama3?tab=readme-ov-file`. In this paper, we use Meta Llama 3, which is governed by the Meta Llama 3 Community License Agreement (Meta Platforms, Inc., released April 18, 2024). This tool is employed strictly in accordance with the terms set forth in the licensing agreement and Meta's Acceptable Use Policy.

Raw Data and LLMs-as-Predictors: `https://github.com/CurryTang/Graph-LLM`. (MIT License).

GIA-HAO: `https://github.com/LFhase/GIA-HAO`. (MIT License)

# C  Device Information

All our experiments are conducted on a machine with an NVIDIA A100-SXM4 (80GB memory), Intel Xeon CPU (2.30 GHz), and 512GB of RAM.

# D  More Related Works

**Graph Injection Attacks.** Graph Injection Attacks aim to decrease the performance of GNNs by injecting suspicious nodes into the original graph. The newly injected nodes are allowed to process fake features and form connections with existing nodes on the graph. Wang et al. [29] first attempt to inject fake nodes to degrade the performance of GNNs. NIPA [25] applies reinforcement learning to make graph injection decisions. AFGSM [28] uses an approximation strategy to linear the model and solve the objective function efficiently. TDGIA [36] selects edges for newly added edges based on a topological defective edge selection strategy. It then generates node features by optimizing a proposed feature smooth objective function. Chen et al. [2] add a regularization term to the existing objective function for GIA to improve homophily unnoticeability. GANI [4] employs a statistics-based approach to calculate node features, ensuring the generated features remain unnoticeable but leading to a significant decline in performance. All the GIAs mentioned are designed at the embedding level. Despite some constraints being applied to the injected features, the semantic coherence and logical consistency of generated features at the text level remain unexplored.

**Graph Modification Attacks.** Graph Modification Attacks aim to decrease the performance of GNNs by modifying the graph structure or node features of the original graphs. Nettack [37] adopts a greedy strategy to select edge and node feature perturbations with unnoticeability constraints. Zügner et al. [38] treat the graph structure as a hyperparameter and use meta-gradients to solve the bilevel poisoning attack objective. Xu et al. [31] use Projected Gradient Descent (PGD) to find the best perturbation matrix. The above algorithms significantly degrade the performance of GNNs, yet all require $O(N^2)$ space complexity. To address the scalability issues, Geisler et al. [5] propose PRBCD and GRBCD using Randomized Block Coordinate Descent. The algorithms are of $O(|\mathcal{E}|)$ time and space complexity and can be scaled up to datasets containing more than 1 million nodes.

**Raw Text learning on TAGs.** With the development of LLMs, works combining them with GNNs have continuously emerged, showing outstanding performance in various graph-related tasks on

TAGs. TAPE [9] uses LLMs as a feature enhancer to obtain high-quality node features. In Instruct-GLM [32], the authors propose scalable prompts based on natural language instructions that describe the topological structure of a graph. Guo et al. [7] analyze the ability of LLMs to understand graph data and solve graph-related tasks. Chen et al. [3] conduct a benchmark evaluating the performance of LLMs as the feature enhancer and the predictor, respectively. More related works about LLMs in Graph could be referred to [15, 24, 14]. Although LLMs have seen numerous applications in graph tasks, works exploring the robustness of GNNs are still sparse.

Although Graph Modification Attacks are powerful attacks, recent works point out that they do not align well with real-world attack scenarios [33], and they are theoretically inferior to GIAs in attack performance [2]. Moreover, while raw texts are proven to be helpful in improving the performance of graph tasks, works exploring the robustness of GNNs at the text level are still sparse.

## E  Datatset Statistics and Budgets of Attacks

Dataset information is summarized in Table 5 and Table 6. The budget of GIAs is summarized in Table 7.

Table 5: Statistics of datasets. Edges (UnD.) stands for the number of edges after transforming each dataset into undirected. Avg.(Max) Sparsity stands for the average (max) percentage of non-zero elements in BoW embeddings of the raw texts.

| Dataset | Nodes | Edges (UnD.) | Classes | Avg. Degree | Avg. Sparsity | Max Sparsity |
|---------|-------|--------------|---------|-------------|---------------|--------------|
| Cora | 2,708 | 10,556 | 7 | 3.90 | 0.06 | 0.17 |
| CiteSeer | 3,186 | 8,450 | 6 | 2.65 | 0.06 | 0.13 |
| PubMed | 19,717 | 88,648 | 3 | 4.50 | 0.10 | 0.23 |
| ogbn-arxiv | 169,343 | 2,332,486 | 40 | 13.77 | 0.06 | 0.33 |
| Reddit | 33,434 | 396,896 | 2 | 11.87 | 0.04 | 0.44 |

Table 6: The statistics reflect the number of words used in each dataset. "Avg." represents the average word count per dataset, while "Std." indicates the standard deviation of word counts. Assuming a normal distribution of word lengths, "Avg. + 2Std." denotes the upper bound of the 95% confidence interval. This metric is employed as a benchmark for establishing maximum word/token limits in paper generation experiments.

| Dataset | Min | Max | Avg. | Std. | Avg. + 2Std. |
|---------|-----|-----|------|------|--------------|
| Cora | 2 | 757 | 132.59 | 83.60 | 299.79 |
| CiteSeer | 4 | 397 | 150.85 | 45.58 | 242.01 |
| PubMed | 2 | 889 | 241.18 | 71.68 | 384.54 |
| ogbn-arxiv | 15 | 1,410 | 170.36 | 57.24 | 284.84 |
| Reddit | 3 | 3,708 | 134.34 | 165.61 | 465.56 |

Table 7: Attack budgets on the datasets.

| Dataset | Nodes | Degree | Node Per.(%) | Edge Per. (%) |
|---------|-------|--------|--------------|---------------|
| Cora | 60 | 20 | 2.22 | 22.74 |
| CiteSeer | 90 | 10 | 2.82 | 21.30 |
| PubMed | 400 | 25 | 2.03 | 22.56 |
| ogbn-arxiv | 1500 | 100 | 0.89 | 12.86 |
| Reddit | 500 | 30 | 1.50 | 7.56 |

## F  Further Illustration and Proof to Theorem 1

The formulation for Theorem 1 relies on the below intuitions.

- **Performance:** An attack is more effective against node $v_t$ if it uses more words $w_n \in W_n$.

- **Unnoticeability:** An attack is less detectable if the similarity between $x_t$ and $x_i$ is higher.

- **Interpretability:** An attack is more interpretable if fewer words are required to use, where the $x_t \cdot \mathbf{1}$ is smaller.

In the intuition behind performance, we actually believe that words beneficial to the classification of the target node are located in $W_u$, while words detrimental to the classification of the target node are located in $W_n$. However, strictly speaking, not all words in $W_n$ have a negative impact on the classification of the target node. Additionally, when a certain amount of classification words is introduced, the negative impact becomes marginal and does not consistently increase. Therefore, using more words $w_n \in W_n$ does not linearly correspond to higher performance.

However, this does not affect the insight provided by Theorem 1. Since $k \ll F$, the continuous increase in the usage of $w_n \in W_n$ within a certain range is significant. Since we do not consider situations where $m \gg k$, we indeed focus on "uses more words $w_n \in W_n$ in a certain range" intrinsically. Thus, under this premise, the theorem's description of performance remains applicable.

*Proof.* The problem is formulated as:

$$\max \quad b \tag{4}$$
$$\text{s.t.} \quad a + b \leq m, \tag{5}$$
$$\frac{a}{\sqrt{k}\sqrt{a+b}} \geq c. \tag{6}$$

From inequality (6), we obtain:

$$a^2 - c^2 k a - c^2 k b \geq 0,$$

which leads to:

$$a \geq \frac{c\left(ck + \sqrt{k(4b + c^2 k)}\right)}{2} \quad \text{or} \quad a \leq \frac{c\left(ck - \sqrt{k(4b + c^2 k)}\right)}{2} < 0.$$

Since $a$ is positive, the only feasible solution is:

$$a \geq \frac{c\left(ck + \sqrt{k(4b + c^2 k)}\right)}{2}.$$

By substituting this into equation (5), we obtain:

$$b \leq m - a \leq m - \frac{c\left(ck + \sqrt{k(4b + c^2 k)}\right)}{2}.$$

Solving the resulting inequality with respect to $b$, we find:

$$b \leq m - c\sqrt{mk}.$$

Hence, the solution to the maximization problem is:

$$\max(b) = \left\lfloor m - c\sqrt{mk} \right\rfloor.$$

$\square$

## G  Collections of Illustrations

### G.1  Illustration of Interpretable Regions

The illustration of interpretable regions is provided in Figure 6.

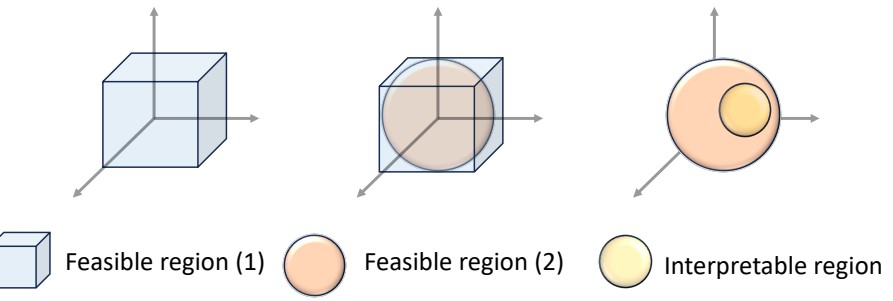

Figure 6: Illustration of the interpretable regions. **Left**: Feasible solution set under constraints (3). **Middle**: The theoretically feasible set of PLM embeddings. **Right**: Subset of embeddings corresponding to interpretable texts, representing a narrower selection within the feasible sets.

## G.2 Alternative for PGD in ITGIA

In addition to the methods mentioned in the main body of the paper, we also considered using the Riemannian Gradient Descent as a way to obtain the embedding for ITGIA. This method performs gradient descent in the tangent space of the sphere to ensure the resulting embedding's normality.

Formally, to solve the optimization problem

$$\min_{x} \quad f(x)$$
$$\text{subject to} \quad \|x\| = 1, \tag{7}$$

using Riemannian gradient descent, we follow these steps:

**Initialization**: Start with an initial point $x_0$ such that $\|x_0\| = 1$.

**Gradient Calculation**: At each iteration $k$, compute the Euclidean gradient $\nabla f(x_k)$.

**Projection onto Tangent Space**: Project the Euclidean gradient onto the tangent space of the sphere at $x_k$:
$$\nabla_R f(x_k) = \nabla f(x_k) - \langle \nabla f(x_k), x_k \rangle x_k, \tag{8}$$
where $\nabla_R f(x_k)$ is the Riemannian gradient.

**Update Step**: Move in the direction of the Riemannian gradient and re-project onto the sphere:
$$x_{k+1} = \text{Proj}_{\mathbb{S}}(x_k - \alpha_k \nabla_R f(x_k)), \tag{9}$$
where $\alpha_k$ is the step size and $\text{Proj}_{\mathbb{S}}(y) = \frac{y}{\|y\|}$ ensures that the updated point lies on the sphere.

**Iteration**: Repeat steps 2-4 until convergence.

The PGD method we adopted can be regarded as the variant of Riemannian Gradient Descent without the "Projection onto Tangent Space" step. Although the Riemannian Gradient Descent has better theoretical properties [1], since we did not observe significant advantages over PGD in our experiments, we opted for the simplicity of using PGD, as discussed in the main body of our paper.

## G.3 Illustration of Co-occurrence Constraint

The co-occurrence constraint in [28] requires that columns that do not appear together in the original dataset must not appear together in the injected embeddings either.

Let $\mathbf{X}$ be a binary feature matrix with dimensions $N \times F$, where $N$ represents the number of samples and $F$ represents the number of features. Each element $x_{ij}$ in the matrix $\mathbf{X}$ is defined as:

$$x_{ij} = \begin{cases} 1 & \text{if word } j \text{ is present in text} i, \\ 0 & \text{otherwise.} \end{cases}$$

The co-occurrence matrix $\mathbf{C}$ is then an $F \times F$ matrix where each element $c_{ij}$, representing the pair of words $i$ and $j$, is computed as follows:

Table 8: The sparsity of co-occurrence matrix.

| Dataset | Cora | CiteSeer | PubMed |
|---|---|---|---|
| Sparsity (%) | 96.99 | 98.21 | 99.97 |

$$\mathbf{C} = (\mathbf{X}^T\mathbf{X}) > 0$$

Here, $\mathbf{X}^T$ is the transpose of $\mathbf{X}$, and the multiplication $\mathbf{X}^T\mathbf{X}$ results in a matrix where each element $c'_{ij}$ is the sum of the products of corresponding elements of $\mathbf{X}$:

$$c_{ij} = \sum_{k=1}^{n} x_{ki}x_{kj}$$

The value $c_{ij}$ represents the number of texts where both words $i$ and $j$ are present. The physical meaning of $c_{ij} > 1$ is that features $i$ and $j$ co-occur in at least one text, indicating a possible association or dependency between them within the dataset. A co-occurrence-constrained FGSM only allows flips that satisfy if word $i$ and word $j$ appear in the injected embeddings, they must have $c_{ij} > 1$.

We examine the sparsity of $\mathbf{C}$ in Table 8. We can see that almost every two words have co-occurred before, making the constraints meaningless.

## H Experiment Details

### H.1 Setup and Hyperparameters

**Attacks.** For the injected embedding features, we use PGD [33, 2] with a learning rate of 0.01, and the training epoch is 500. An early stop of 100 epochs according to the accuracy of the surrogate model on the target nodes. For TDGIA and ATDGIA, we use the method of [36] to generate injected embeddings. The weights $k_1$ and $k_2$ are set 0.9 and 0.1 by default. The sequential step is set as 1.0 for MetaGIA to save computational costs, while 0.20 for methods. For Binary FGSM, the batch size is set to as $B = 1$ in Cora and CiteSeer, and $B = 50$ for PubMed. For the hyperparameters of SeqGIA, MetaGIA, and SeqAGIA, we directly follow the setting as in [2]. A simplified description of the Sequential injection framework is provided in Appendix 1, and the FGSM function is provided in Appendix 2.

**Harmonious Adversarial Objective.** The HAO constraint can be formulated as:

$$\mathcal{L} = \mathcal{L}_{pred} + \gamma\mathcal{L}_{HAO}.$$

Here, $\mathcal{L}_{pred}$ represents the cross-entropy loss on target nodes, calculated between the current predictions of the surrogate model and the original predictions made on the clean dataset by the same model. The loss term $\mathcal{L}_{HAO} = sim(\mathbf{X}'_{\text{inj}}, (\mathbf{X}'\mathbf{A}')_{\text{inj}})$, where $sim$ returns the cosine similarity between injected node features and propagated injected node features. We set $\gamma = 10$ in SeqGIA, SeqAGIA and MetaGIA, $\gamma = 1$ in TDGIA, ATDGIA and set $\gamma = 1e-6$ in FSGM, where we empirically find a balance in the attack performance and unnoticeability to escape from homophily-based defender.

Note that in Figure 2, $\gamma$ is multiplyed by the weights as the final coefficient. Denote $\gamma' = \gamma * w$, where $w$ is the weight listed in the x-axis of the figure, we apply $\mathcal{L} = \mathcal{L}_{pred} + \gamma'\mathcal{L}_{HAO}$ as the final loss function.

**Defense.** For both GCN and EGNNGuard, we set the number of layers as 3 and use a hidden dimension of 64. We set the dropout rate as 0.5 and adopt Adam optimizer [12] with a learning rate of 0.01. The number of training epochs is 400. We adopt the early stop strategy with a patience of 100 epochs. The threshold is set as 0.1 following [2], which we find powerful enough in detecting noticeable injections.

**Others.** The temperature of the GPT-turbo is set as 0.7. In ITGIA, the beam-width is set as 2. In WTGIA, the max-token for gpt-turbo and Llama3 is set to 500 for Cora and CiteSeer and 550 for

PubMed. We set the the number of dialogs in multi-round corrections as 3. We set the max-token to avoid over-length generation. Based on the typical ratio of 0.75:1 between the number of words and the number of tokens, we proportionally scale this relationship using the data from Table 6 to obtain the specific numbers for this limitation.

## H.2 Prompt Design

Table 9 provides our prompt designs for different prompt strategies of vanilla text-based GIAs. The <adj/example_paper_content> is comprised of paper id, title, and abstract of adjacent/example nodes, and ... denotes fillable blanks for the model to fill in. The <LMIN> and <LMAX> correspond to the length limit for each dataset as specified in Table 6. Generally, our experiments find that the current instructions allow the LLM to produce output that conforms well to the expected format without significant deviations.

Table 9: Prompts of vanilla text-based GIAs. Number of positive examples are set to as 5, and number of negative examples are set to as 3.

| Attack | Prompts |
|---|---|
| Vanilla Heterophily | Task: Paper Generation. There are <num_classes> types of paper, which are <category_names>. Positive Examples of the papers are: Content: <example_paper_content> Type: <cur_label>: Negative Examples of the papers are: Content: <example_paper_content> Generate a title and an abstract for paper <inj_node> which is dissimilar to the negative examples, but belongs to at least one of the types in <category_names> similar to positive examples. Length limit: Min: <LMIN> words, Max: <LMAX> words. Title: ..., Abstract: ... |
| Vanilla Mixing | Task: Paper Generation. There are <num_classes> types of paper, which are <category_names>. Examples of the papers are: Content: Generate a title and an abstract for paper <inj_node> that belong to all the above paper types. The generated content should be able to be classified into any type of paper. Length limit: Min: <LMIN> words, Max: <LMAX> words. Title: ..., Abstract: ... |
| Vanilla Random | Task: Paper Generation. There are <num_classes> types of paper, which are <category_names>. Generate a title and an abstract for paper <inj_node> that does not belong to any one of the paper types. For example, the paper could belong to ['history', 'art', 'philosophy', 'sports', 'music'] and types like that. Length limit: Min: <LMIN> words, Max: <LMAX> words. Title: ..., Abstract: ... |

Table 10 provides prompts of WTGIA with different generation type.

# I Collections of Pseudo-code

## I.1 Pseudo-code for Sequential GIAs

The pseudo-code for sequential node injection strategy is displayed in Algorithm 1. A simple illustration of the feature update function of SeqGIA, MetaGIA, TDGIA, and SeqAGIA is given in Algorithm 3. The pseudo-code for Binary FGSM feature update functions is displayed in Algorithm 2. For FGSM binary attacks, we implement the structure updates in the sequential node injection manner, as it is shown more powerful by previous works [28, 2]. The initialization of $\mathbf{X}_{inj}$ is set as $\mathbf{X}_{inj} = \mathbf{0}$

Table 10: Prompts of vanilla text-based GIAs.

| Generation Type | Prompts |
|---|---|
| Llama/GPT | Generate a title and an abstract for an academic article.
Ensure the generated content explicitly contains the following words: <use_words>.
These words should appear as specified, without using synonyms, plural forms, or other variants.
Length limit: <max_words> words.
Output the TITLE and ABSTRACT without explanation.
TITLE:...
ABSTRACT:... |
| Llama/GPT-topic | There are <num_classes> types of paper, which are <category_names>.
Generate a title and an abstract for paper belongs to one of the given categories.
Ensure the generated content explicitly contains the following words: <use_words>.
These words should appear as specified, without using synonyms, plural forms, or other variants.
Length limit: <max_words> words.
Output the TITLE and ABSTRACT without explanation.
TITLE:...
ABSTRACT:... |

and applying feature update function as in Algorithm 2. For the details about the structure update function $\mathcal{H}$ of SeqGIA, MetaGIA, TDGIA, ATDGIA, AGIA, and the feature update function of TDGIA and ATDGIA, please refer to [2, 36].

Note that in ITGIA, if normalized, after each single epoch of feature update, we project the output $\mathbf{X}_{\text{inj}}$ to a unit sphere. A text illustration of PGD can be found in G.2, where an alternative to vanilla PGD is discussed.

---

**Algorithm 1:** Sequantial Attack Framework

---

**Input** : Graph $\mathcal{G} = (\mathcal{V}, \mathcal{E}, \mathbf{X})$, Adjacency matrix $\mathbf{A}$, Number of features $F$, Number of injected nodes $N_{\text{inj}}$, Number of target nodes $|\mathcal{V}_T|$, Surrogate model $f$, Feature update function $\mathcal{F}$, Structure update function $\mathcal{H}$; (Function specific arguments excluded)

**Output** : Attacked Graph $\mathcal{G}' = (\mathcal{V}', \mathcal{E}', \mathbf{X}')$

**Parameter** : Sequential step $st$, Normalization flag $Norm$;

1 $\mathcal{V}' \leftarrow \mathcal{V}; \mathcal{E}' \leftarrow \mathcal{E}; \mathbf{X}' \leftarrow \mathbf{X}$
2 $n_{inj} \leftarrow 0$
3 $Y_{orig} = f(\mathbf{X}, \mathbf{A})$
4 **while** $n_{inj} < N_{inj}$ **do**
5      $n \leftarrow \min(N_{\text{inj}} - N_{\text{Total}}, \lfloor N_{\text{inj}} * st \rfloor)$
6      Initialize $\mathcal{V}_{\text{inj}}, \mathcal{E}_{\text{inj}}, \mathbf{X}_{\text{inj}}$          // Initialization of the current epoch
7      $\mathcal{V}' \leftarrow \mathcal{V}' \cup \mathcal{V}_{\text{inj}}; \mathcal{E}' \leftarrow \mathcal{E}' \cup \mathcal{E}_{\text{inj}}$
8      $\mathcal{V}_{\text{inj}}, \mathcal{E}_{\text{inj}} \leftarrow \mathcal{H}(\mathbf{X}, \mathbf{X}_{\text{inj}}, \mathcal{V}', \mathcal{E}', f, Y_{orig})$          // Update Structure
9      $\mathcal{V}' \leftarrow \mathcal{V}' \cup \mathcal{V}_{\text{inj}}; \mathcal{E}' \leftarrow \mathcal{E}' \cup \mathcal{E}_{\text{inj}}$
10      $\mathbf{X}_{\text{inj}} \leftarrow \mathcal{F}(\mathbf{X}, \mathbf{X}_{\text{inj}}, \mathcal{V}', \mathcal{E}', f, Y_{orig}, Norm)$          // Update Features
11      $\mathbf{X}' = \mathbf{X}' \cup \mathbf{X}_{\text{inj}}$
12      $n_{inj} \leftarrow n_{inj} + n$
13 **return** *Attacked Graph* $\mathcal{G}' = (\mathcal{V}', \mathcal{E}', \mathbf{X}')$

---

---

**Algorithm 2:** Feature Update for Continuous Features

---
**Input**      :Model $f$, features $\mathbf{X}$, attack features $\mathbf{X}_{inj}$, adjacency matrix $A'$, original labels $Y_{orig}$, target indices $T$, epochs $e$, learning rate $\eta$;

**Output**   :Updated attack features $\mathbf{X}_{inj}$

---
1  **for** $epoch \leftarrow 0$ **to** $e$ **do**
2     $\mathbf{X}_{\text{cat}} = \mathbf{X} \cup \mathbf{X}_{inj}$
3     $\mathbf{X}_{inj} \leftarrow \text{Proj}\left(\mathbf{X}_{inj} - \eta\nabla_{\mathbf{X}_{inj}}\mathcal{L}_{ch}\left(f(\mathbf{X}_{\text{cat}}, \mathbf{A}'), Y_{orig}\right)\right)$     `// Project to unit sphere for ITGIA`
4  **return** $X_{inj}$

---

---

**Algorithm 3:** Fast Gradient Sign Method (FGSM) Update for Binary Features

---
**Input**      :Model $f$, features $\mathbf{X}$, attack features $\mathbf{X}_{inj}$, adjacency matrix $\mathbf{A}'$, original labels $Y_{orig}$, target indices $T$, sparsity budget $S$, batch size $B$

**Output**   :Updated attack features $\mathbf{X}_{inj}$

---
1  $F \leftarrow \mathbf{X}.\text{shape}[1]; F' \leftarrow \lfloor S \cdot D \rfloor$         `// Allowed flips per node.`
2  **while** *any row flips* $< F'$ **do**
3     $\mathbf{X}_{cat} \leftarrow \mathbf{X} \cup \mathbf{X}_{inj}$
4     $Z \leftarrow f(\mathbf{X}_{cat}, \mathbf{A}')$
5     $\nabla\mathcal{L} \leftarrow \nabla_{\mathbf{X}_{inj}}\mathcal{L}(Z, Y_{orig}, T)$
6     Determine flip mask $M \leftarrow$ (Current flips per row $< F'$)
7     $G_{valid} \leftarrow \nabla\mathcal{L} \cdot M$        `// Mask gradients by rows that can still flip`
8     Compute flip direction $G_{flip} \leftarrow \text{sign}(G_{valid}) - (\mathbf{X}_{inj} == 1)$
9     Select indices of top $B$ gradients $I \leftarrow$ Top-K indices from $G_{flip}$
10    **foreach** $i \in I$ **do**
11       $r, c \leftarrow$ Row and column of index $i$
12       **if** *Row $r$ can still flip* **then**
13          Flip $X_{inj}[r, c] \leftarrow 1 - \mathbf{X}_{inj}[r, c]$
14          Update the flip counter for row $r$
15 **return** $\mathbf{X}_{inj}$

---

### I.2 Pseudo-code for Vanilla Text-Level GIAs

The pseudo-code for vanilla text-level GIAs is displayed in Algorithm 4, which is implemented in a sequential manner following SeqGIA. Note that the feature is not updated within the "while loop" since gradient information is infeasible.

## J   Interpretability of Generated Texts

### J.1  Perplexity and Use Rate

We calculate perplexity based on GPT2 [21] following `https://huggingface.co/docs/transformers/perplexity`. Note that due to the model capacity constraint, the output of IT-GIA is truncated to 32 tokens. So, we truncate all texts into 32 tokens before calculating Perplexity to make them comparable to each other. We present the results of average perplexity, and use rate in WTGIA in Table 11, Table 12, Table 13 and Table 14.

In the experiments, VTGIA consistently achieves the lowest perplexity scores, indicating superior performance, followed by the WTGIA model. Conversely, ITGIA exhibits the highest perplexity, rendering it the least interpretable. Notably, even the variant incorporating -HAO fails to enhance interpretability corrsponding to normal texts. Furthermore, our findings suggest a positive correlation between model use rate and perplexity. Specifically, a lower use rate corresponds to reduced perplexity, thereby enhancing interpretability. This observation aligns with the hypothesis that as the use rate decreases, LLMs preferentially select words from a specified set that are more interpretable, rather than striving to utilize all specified words.

**Algorithm 4:** Text to Graph Attack

| | |
|---|---|
| **Input** | :Graph $\mathcal{G} = (\mathcal{V}, \mathcal{E}, \{s_i\})$, Class set $C$, Number of injected nodes $N_{\text{inj}}$, Number of target nodes $|\mathcal{V}_T|$. |
| **Output** | :Attacked Graph $\mathcal{G}' = (\mathcal{V}', \mathcal{E}', \{s_i\}')$ |
| **Parameter** | :Sequential step $st$, prompt type $p$ |

**1** $\mathcal{V}' \leftarrow \mathcal{V}; \mathcal{E}' \leftarrow \mathcal{E}$
**2** **if** $p ==$ *Heterophily prompt* **then**
    /* Integrate attacked graph into prompts */
**3**    $\mathcal{V}', \mathcal{E}' \leftarrow$ Embedding-based GIAs
**4**    $\{s_i\}_{\text{inj}} \leftarrow$ GenerateText($N_{\text{inj}}, , \mathcal{G}', \mathcal{V}', \mathcal{E}'$)
**5** **else**
**6**    $\{s_i\}_{\text{inj}} \leftarrow$ GenerateText($N_{\text{inj}}, , \mathcal{G}'$)
**7** $\{s_i\}' = \{s_i\} \cup \{s_i\}_{\text{inj}}$
**8** $\mathbf{X}' \leftarrow$ TextEmbedding($\{s_i\}_{\text{inj}}$)
**9** $N_{\text{Total}} \leftarrow 0$
    /* Graph structure refinement */
**10** **while** $N_{Total} < N_{inj}$ **do**
**11**    $n \leftarrow \min(N_{\text{inj}} - N_{\text{Total}}, \lfloor N_{\text{inj}} * st \rfloor)$
**12**    $\mathcal{V}', \mathcal{E}' \leftarrow$ AdaptiveInjection($\mathcal{V}', \mathcal{E}', \mathbf{X}'$)
**13**    $N_{\text{Total}} \leftarrow N_{\text{Total}} + n$
**14**
**15** **return** *Attacked Graph* $\mathcal{G}' = (\mathcal{V}', \mathcal{E}', \{s_i\}')$

Table 11: Average perplexity of raw text generated by VTGIA and ITGIA. Clean refers to the average perplexity of original dataset.

| Dataset | Clean | VTGIA-Het. | VTGIA-Rand. | VTGIA-Mix. | ITGIA | ITGIA-HAO |
|---|---|---|---|---|---|---|
| Cora | 110.47 | 14.02 | 18.12 | 16.63 | 623.65 | 546.89 |
| CiteSeer | 66.71 | 14.37 | 16.53 | 21.21 | 705.41 | 379.80 |
| PubMed | 30.85 | 8.21 | 16.76 | 13.52 | 503.14 | 348.07 |

Table 12: Average perplexity ($\downarrow$) and use rate of raw texts generated by WTGIA w.r.t sparsity budget on Cora dataset.

| WTGIA Variant | Avg. | 0.10 | 0.15 | 0.20 |
|---|---|---|---|---|
| GPT Perplexity | 53.88 | 43.11 | 39.08 | 35.60 |
| GPT Use Rate (%) | 84.29 | 80.84 | 66.48 | 52.72 |
| GPT-Topic Perplexity | 30.70 | 26.92 | 26.40 | 25.01 |
| GPT-Topic Use Rate (%) | 81.78 | 73.93 | 58.85 | 44.81 |
| Llama Perplexity | 90.23 | 75.95 | 58.17 | 54.73 |
| Llama Use Rate (%) | 93.43 | 86.71 | 54.60 | 51.29 |
| Llama-Topic Perplexity | 83.21 | 65.97 | 54.60 | 55.69 |
| Llama-Topic Use Rate (%) | 93.08 | 86.03 | 71.56 | 50.67 |

Table 13: Average perplexity (↓) and use rate of raw texts generated by WTGIA w.r.t sparsity budget on CiteSeer dataset.

| WTGIA Variant | Avg. | 0.10 | 0.15 | 0.20 |
|---|---|---|---|---|
| GPT Perplexity | 60.55 | 51.30 | 43.28 | 38.09 |
| GPT Use Rate (%) | 87.54 | 79.79 | 65.27 | 53.20 |
| GPT-Topic Perplexity | 38.34 | 36.39 | 30.59 | 38.34 |
| GPT-Topic Use Rate (%) | 83.11 | 72.91 | 55.98 | 45.57 |
| Llama Perplexity | 86.11 | 67.57 | 51.89 | 48.79 |
| Llama Use Rate (%) | 96.69 | 87.72 | 59.75 | 59.73 |
| Llama-Topic Perplexity | 83.52 | 65.02 | 44.23 | 51.06 |
| Llama-Topic Use Rate (%) | 96.58 | 88.18 | 73.60 | 54.45 |

Table 14: Average perplexity (↓) and use rate of raw texts generated by WTGIA w.r.t sparsity budget on PubMed dataset.

| WTGIA Variant | Avg | 0.15 | 0.20 |
|---|---|---|---|
| GPT Perplexity | 45.57 | 42.65 | 78.08 |
| GPT Use Rate (%) | 73.06 | 58.90 | 40.83 |
| GPT-Topic Perplexity | 30.78 | - | - |
| GPT-Topic Use Rate (%) | 59.72 | - | - |
| Llama Perplexity | 55.89 | 43.84 | 40.24 |
| Llama Use Rate (%) | 82.23 | 72.60 | 61.97 |
| Llama-Topic Perplexity | 58.63 | 44.22 | 40.02 |
| Llama-Topic Use Rate (%) | 82.33 | 72.21 | 62.20 |

### J.2 Examples of VTGIA

We show the generated examples on the Cora dataset for VTGIA, one for each prompt type.

**Heterophily Prompt Example.** Title: Reinforcement Learning Techniques for Rule Learning Optimization Abstract: This paper explores the application of reinforcement learning techniques to optimize the process of rule learning. Traditional rule-learning algorithms often struggle with complex datasets and high-dimensional feature spaces, leading to suboptimal rule sets. By leveraging the power of reinforcement learning, we propose a novel approach that dynamically adapts the rule learning process based on feedback from the environment. We start by formulating the rule learning problem as a Markov decision process, where the agent's actions are the selection and refinement of rules based on the input data. We introduce a reward function that incentivizes the discovery of rules with high predictive power while maintaining simplicity and interpretability. Through a series of experiments on benchmark datasets, we demonstrate the effectiveness of our approach in outperforming traditional rule learning methods. Furthermore, we investigate the convergence properties of our reinforcement learning algorithm and provide theoretical bounds on its performance. Our results show that by combining reinforcement learning with rule learning, we can achieve more accurate and concise rule sets, leading to improved generalization performance on unseen data. Overall, this paper contributes to the intersection of reinforcement learning and rule learning, opening up new possibilities for optimizing rule-based systems in various domains.

**Random Prompt Example.** Title: Exploring the Influence of Martial Arts on Japanese History Abstract: This paper delves into the deep-rooted connection between martial arts and the historical landscape of Japan. By tracing the evolution of various martial art forms such as Judo, Kendo, and Karate, we analyze how these disciplines have not only shaped the physical combat techniques but also contributed significantly to the cultural, philosophical, and political development of Japan. Through studying the impact of samurai warriors, martial arts schools (ryu), and renowned martial artists throughout different historical periods, this paper uncovers the intricate interplay between martial arts and Japan's societal norms, values, and power dynamics. Furthermore, by exploring the

ways in which martial arts have been depicted in art, literature, and popular culture over the centuries, we gain a deeper understanding of how these practices have transcended mere combat techniques to embody spiritual, ethical, and aesthetic principles. Ultimately, this paper aims to shed light on the enduring legacy of martial arts in Japanese history and the profound influence they continue to exert on various aspects of Japanese society today.

**Mixing Prompt Example.** Title: Unified Approaches in Machine Learning and Control Systems Abstract: This paper presents a novel approach to machine learning and control systems by integrating various techniques such as probabilistic methods, neural networks, case-based reasoning, genetic algorithms, theory, reinforcement learning, and rule learning. The proposed framework leverages the strengths of each approach to address complex problems in a unified manner. It introduces a new algorithm, contextual machine learning, which incorporates adaptative history-sensitive models for source separation, making use of the temporal structure of input data. Furthermore, the paper explores the application of non-linear systems in robust control and the use of functional programming by analogy in the domain of program synthesis. Additionally, it presents a learning algorithm for Disjunctive Normal Form (DNF) under the uniform distribution, with implications for the case when certain parameters are constant. The paper also discusses the use of Gaussian noise in sigmoidal belief networks and the training of such networks using slice sampling for inference and learning. Finally, it showcases the application of radial basis function networks in predicting power system security margin. Overall, this paper provides a comprehensive overview of the diverse applications and advancements in machine learning and control systems, demonstrating the potential for creating unified approaches that transcend traditional boundaries.

## J.3 Examples of ITGIA

We show the generated examples on the Cora dataset for ITGIA in this subsection. The output is highly uninterpretable due to ill-defined interpretable regions for injected embeddings.

**ITGIA-TDGIA without HAO, Cos: 0.136**

- of the following: MC Dallas' NSA sign, and some of the relays of the past few years: Trumpet Jam's sinus
- the liner notes of The MC6's "Desirty Pigs": the relayed that some of the plaque
- liner notes of The MC's "The Lion's Pie" album: some relayed that the 'bumpy' fluid
- the following: One of the croons of the "Petsy on the MC12" relayed: Demitab

**ITGIA-TDGIA with HAO, Cos: 0.742**

- "Sympteric" Alignment. This is one of the best examples of how to achieve high level information retrieval algorithms using proportion
- "Asymptotically" by Grant. This is a key method for constructing local information representation systems that support multicluster connections
- Emilia. The "Proplective information with no associativity at all" NNNN algorithms are based on Monte simple scenarios

## J.4 WTGIA

We show the generated examples on the Cora dataset with sparsity budget set as "average" for WTGIA in the subsection. We choose the FGSM-TDGIA for embedding generation. We display one example for each generation type.

**GPT.** Title: Hierarchical Recurrent Neural Networks for Visual Sequence Identification and Dimensional Improvement

Abstract: In this article, we demonstrate the idea of using hierarchical recurrent neural networks for visual sequence identification, resulting in dimensional improvement. We show that this approach requires the underlying mechanisms of backpropagation through time to obtain a more powerful representation of visual sequences. Additionally, we identify the nonlinear nature of the sequence identification problem and how the use of recurrent neural networks leads to nonlinear improvement in identifying visual sequences. Furthermore, we present a novel approach to dealing with hierarchical

structures in visual sequences, utilizing the power of recurrent neural networks to identify objects in a hierarchical and sample-efficient manner. This work contributes to the science of visual sequence processing and demonstrates the capability of recurrent neural networks to handle complex visual sequences, such as trees, lines, and vector objects.

**GPT-Topic.** Title: Hierarchical Neural Networks for Visual Sequence Identification: Exploring the Power of Recurrent Mechanisms in Computer Science

Abstract: This paper presents an investigation into the use of hierarchical neural networks for visual sequence identification in computer science. The underlying idea is to demonstrate the power of recurrent mechanisms in identifying dimensional visual sequences. The resulting improvement in identification accuracy leads to the direct application of backpropagation mechanisms for obtaining nonlinear vector representations of visual sequences. In addition, the paper explores the application of hierarchical neural networks in identifying hierarchical tree structures within visual sequences. The research also requires the exploration of probabilistic methods for identifying sample sequences and tree structures within visual sequences. This work contributes to the advancement of science in computer vision and object recognition by exploring the hierarchical structure of sequences.

**Llama.** Title: Harnessing the Power of Hierarchical Neural Trees for Dimensionally-Transformed Sequence Identification and Improvement through Addition and Recurrent Back-Propagation

Abstract: This article proposes a groundbreaking idea that combines the strengths of hierarchical neural trees and recurrent back-propagation to identify and classify nonlinear sequences. By leveraging the underlying mechanisms of neural trees and incorporating the addition of recurrent feedback, we demonstrated a remarkable improvement over conventional sequence identification methodologies through the power of back-propagation. Our methodology requires the creation of a vector-based object that represents the sequence, which is then fed into a hierarchical neural line that utilizes recurrent back-propagation to identify the underlying dimensional transformations. This leads to a resulting sample of sequences that can be analyzed and visualized to obtain a deeper comprehension of the object's underlying mechanisms. To validate our idea, we conducted a comprehensive sequence identification experiment, where we utilized a sample of sequences to train and fine-tune our hierarchical neural trees. Our resulting sequences demonstrated a remarkable capability to identify and classify nonlinear sequences, out-performing other leading methodologies, and providing a visual insight into the underlying mechanisms. This breakthrough has the power to revolutionize the science of sequence identification and has far-reaching implications for fields such as signal-processing, image-recognition, and predictive analytics.

**Llama-Topic.** Title: Hierarchical Recurrent Neural Trees for Dimensionally Reduced Sequence Identification and Power System Monitoring

Abstract: This article demonstrates the effectiveness of hierarchical recurrent neural trees (HRNT) for identifying sequences of nonlinear phenomena occurring within power grids. By leveraging the power of back-propagation through trees, HRNT is capable of capturing underlying mechanisms governing the system's nonlinear interactions. Our HRNT-based sequence identification idea requires a hierarchical arrangement of recurrent neural trees, which are then utilized to identify sequences of interest along a dimensional line of power transmission. Notably, our methodology leads to a resulting improvement of up to [X]% over conventional sequence identification schemes, which has been demonstrated through a sample dataset of power system measurements. Furthermore, our HRNT-based system can effectively identify nonlinear phenomena, such as addition of harmonics, and nonlinear interactions between power system objects, leveraging the power of vector calculus to obtain a comprehensive picture. Our visualizations of the resulting sequences and underlying mechanisms have been found to be particularly insightful for power system scientists, contributing to the advancement of the science and providing a valuable addition to the scientific community.

# K  More Experiments about WTGIA and ITGIA

## K.1  Transfer WTGIA to More Datasets

In Tables 15 and Table 16, we included the ogbn-arxiv dataset [10], containing over 160,000 nodes, and a social network dataset, Reddit [11], with more than 30,000 nodes, to evaluate the scalability and generalization of text-level GIAs. We injected 1,500 nodes into ogbn-arxiv, following the budget

specified in [2], and proportionally injected 500 nodes into Reddit. On both datasets, WTGIA maintains use rates above 90% and shows good attack performance when transferred to GTR.

Table 15: Performance of WTGIA with the average sparsity on the ogbn-arxiv dataset. Clean accuracy for BoW embedding is 71.87 ± 0.11, and for GTR embedding is 72.36 ± 0.13.

| LLM Type | Emb. | Random | SeqGIA | TDGIA | ATDGIA | AGIA |
|---|---|---|---|---|---|---|
| Emb. | BoW | 70.25 ± 0.25 | 63.00 ± 3.44 | 60.56 ± 2.69 | 64.33 ± 3.18 | **53.34 ± 1.75** |
| GPT | BoW | 70.19 ± 0.24 | 63.65 ± 2.99 | 60.96 ± 2.33 | 64.53 ± 2.78 | **55.38 ± 1.59** |
| | GTR | 68.44 ± 0.56 | 67.47 ± 0.90 | **64.95 ± 1.18** | 67.24 ± 0.99 | 65.90 ± 0.82 |
| GPT-Topic | BoW | 70.03 ± 0.29 | 65.11 ± 2.53 | 62.47 ± 2.10 | 65.89 ± 2.39 | **56.69 ± 1.45** |
| | GTR | 68.25 ± 0.56 | 67.50 ± 0.88 | **64.97 ± 1.19** | 67.35 ± 0.96 | 65.70 ± 0.80 |
| Llama | BoW | 70.25 ± 0.24 | 63.61 ± 3.33 | 61.09 ± 2.70 | 64.67 ± 3.15 | **53.95 ± 1.74** |
| | GTR | 68.64 ± 0.59 | 67.38 ± 0.91 | **64.72 ± 1.22** | 67.28 ± 1.06 | 65.12 ± 0.97 |
| Llama-Topic | BoW | 70.25 ± 0.24 | 63.67 ± 3.32 | 61.09 ± 2.69 | 64.65 ± 3.14 | **53.98 ± 1.73** |
| | GTR | 68.65 ± 0.58 | 67.24 ± 0.89 | **64.42 ± 1.23** | 67.18 ± 1.05 | 65.45 ± 0.94 |

Table 16: Performance of WTGIA with the average sparsity on the Reddit dataset. Clean accuracy for BoW embedding is 62.84 ± 0.89, and for GTR embedding is 68.28 ± 0.24.

| LLM Type | Emb. | Random | SeqGIA | MetaGIA | TDGIA | ATDGIA | AGIA |
|---|---|---|---|---|---|---|---|
| Emb. | BoW | 59.48 ± 1.36 | 55.34 ± 1.25 | 54.48 ± 0.96 | **50.72 ± 0.78** | 54.65 ± 1.38 | 55.80 ± 1.12 |
| GPT | BoW | 59.84 ± 1.34 | 56.48 ± 1.47 | 55.49 ± 1.21 | **52.03 ± 1.29** | 55.92 ± 1.75 | 56.66 ± 1.29 |
| | GTR | 66.94 ± 0.47 | 66.97 ± 0.47 | 67.13 ± 0.52 | 67.77 ± 0.52 | 67.01 ± 0.52 | **66.87 ± 0.47** |
| Llama | BoW | 59.51 ± 1.35 | 55.48 ± 1.28 | 54.69 ± 1.06 | **50.86 ± 0.87** | 54.71 ± 1.41 | 55.89 ± 1.16 |
| | GTR | 67.39 ± 0.56 | 66.77 ± 0.49 | **66.34 ± 0.66** | 67.02 ± 0.81 | 66.86 ± 0.61 | 66.44 ± 0.51 |

## K.2 Against More Defense Models

In Table 17, Table 18, Table 19, Table 20, Table 21, and Table 22, we include classic methods like GAT [27] and GraphSAGE [8], as well as the EGNNGuard and Layernorm (LN) methods, which have been proven highly effective in [2].

We discover interesting new results, such as LN's effectiveness on text-level GIAs not being as strong as previously reported for embedding-level attacks, especially for WTGIA. This is because embedding-level GIAs often exhibit abnormal norms, which LN exploits for defense. However, text-level GIAs derive embeddings from real text, avoiding structural anomalies and bypassing LN's defenses. This suggests that some traditional defense methods that were particularly effective may be limited to the embedding level. The EGNNGuard method is generally effective and performs well overall. As mentioned in the main paper, attackers can use techniques like HAO to bypass EGNNGuard, but this often involves trade-offs.

Table 17: ITGIA with default HAO weight against more defense models on Cora, GTR embedding.

| Model | Clean | SeqGIA | MetaGIA | TDGIA | ATDGIA | AGIA |
|---|---|---|---|---|---|---|
| GCN | 87.19 ± 0.62 | **65.67 ± 1.48** | 65.77 ± 1.15 | 71.49 ± 1.71 | 74.63 ± 2.48 | 68.59 ± 1.51 |
| GAT | 87.47 ± 0.49 | **70.52 ± 5.90** | 70.77 ± 4.54 | 72.62 ± 4.02 | 74.25 ± 3.64 | 73.89 ± 4.27 |
| SAGE | 85.51 ± 0.66 | **67.38 ± 1.92** | 71.04 ± 1.76 | 74.03 ± 1.95 | 76.31 ± 2.13 | 70.78 ± 2.01 |
| Guard | 87.81 ± 0.36 | 68.83 ± 1.82 | **68.17 ± 1.64** | 73.62 ± 2.24 | 78.33 ± 2.74 | 70.79 ± 1.96 |
| GCN-LNi | 87.07 ± 0.21 | 75.00 ± 0.98 | **73.85 ± 0.69** | 79.96 ± 0.99 | 81.63 ± 1.09 | 76.74 ± 1.32 |

## K.3 Transfer to More Embedding Methods

In Table 23 and Table 24, we included SentenceBert all-MiniLM-L6-v2 [22] as a new embedding backbone. We find that WTGIA shows some transferability between different embeddings, but the

Table 18: WTGIA with average sparsity against more defense models on Cora, BoW embedding.

| Model | Clean | SeqGIA | MetaGIA | TDGIA | ATDGIA | AGIA |
|---|---|---|---|---|---|---|
| GCN | 86.48 ± 0.41 | 48.32 ± 0.74 | 51.58 ± 0.78 | 52.49 ± 1.32 | **35.33 ± 1.29** | 47.81 ± 0.78 |
| GAT | 86.67 ± 0.41 | 51.46 ± 5.08 | 57.65 ± 4.98 | 54.72 ± 7.81 | **33.28 ± 8.31** | 50.17 ± 4.35 |
| SAGE | 85.74 ± 0.57 | 54.86 ± 1.97 | 58.35 ± 1.64 | 58.12 ± 2.10 | 56.01 ± 2.71 | **53.21 ± 1.32** |
| Guard | 84.62 ± 0.38 | 80.58 ± 0.44 | 79.44 ± 0.33 | 81.94 ± 0.51 | **79.33 ± 0.65** | 79.90 ± 0.35 |
| GCN-LNi | 85.70 ± 0.65 | 68.12 ± 1.37 | 69.91 ± 1.49 | 73.97 ± 1.30 | **66.09 ± 2.05** | 67.15 ± 1.79 |

Table 19: ITGIA with default HAO weight against more defense models on CiteSeer, GTR embedding.

| Model | Clean | SeqGIA | MetaGIA | TDGIA | ATDGIA | AGIA |
|---|---|---|---|---|---|---|
| GCN | 75.93 ± 0.41 | **64.79 ± 1.30** | 65.11 ± 1.01 | 67.43 ± 0.89 | 71.89 ± 0.50 | **64.79 ± 1.30** |
| GAT | 75.67 ± 0.53 | **54.96 ± 2.11** | 63.90 ± 2.65 | 61.71 ± 1.43 | 68.80 ± 1.01 | 57.20 ± 1.93 |
| SAGE | 73.84 ± 0.60 | **64.48 ± 1.22** | 66.98 ± 1.17 | 65.50 ± 1.12 | 69.05 ± 0.66 | 65.33 ± 1.32 |
| Guard | 76.84 ± 0.65 | **65.97 ± 1.06** | 67.28 ± 1.62 | 68.77 ± 0.72 | 72.72 ± 0.54 | 66.37 ± 1.04 |
| GCN-LNi | 75.22 ± 0.74 | **65.29 ± 0.81** | 66.97 ± 1.22 | 69.17 ± 1.23 | 71.66 ± 1.03 | 65.71 ± 0.62 |

Table 20: WTGIA with average sparsity against more defense models on CiteSeer, BoW embedding.

| Model | Clean | SeqGIA | MetaGIA | TDGIA | ATDGIA | AGIA |
|---|---|---|---|---|---|---|
| GCN | 75.92 ± 0.55 | **47.52 ± 0.65** | 51.25 ± 0.95 | 49.17 ± 0.94 | 49.50 ± 0.74 | 47.63 ± 0.81 |
| GAT | 76.18 ± 0.60 | 49.38 ± 2.74 | 55.18 ± 4.13 | **48.37 ± 3.96** | 50.12 ± 3.17 | 50.12 ± 2.69 |
| SAGE | 73.54 ± 0.71 | 51.67 ± 1.13 | 55.56 ± 1.19 | **50.29 ± 1.40** | 54.16 ± 1.29 | 51.75 ± 1.07 |
| Guard | 74.71 ± 0.58 | 70.88 ± 0.61 | 72.04 ± 0.53 | 70.40 ± 0.46 | **70.17 ± 0.41** | 71.84 ± 0.47 |
| GCN-LNi | 74.91 ± 0.70 | **52.30 ± 0.81** | 56.46 ± 1.52 | 55.98 ± 1.50 | 58.33 ± 0.91 | 56.48 ± 0.89 |

Table 21: ITGIA with default HAO weight against more defense models on PubMed, GTR embedding.

| Model | Clean | SeqGIA | MetaGIA | TDGIA | ATDGIA | AGIA |
|---|---|---|---|---|---|---|
| GCN | 87.91 ± 0.26 | 66.40 ± 2.33 | **58.56 ± 1.22** | 60.26 ± 1.32 | 76.23 ± 2.08 | 65.77 ± 0.91 |
| GAT | 86.85 ± 0.27 | 62.44 ± 3.96 | 60.40 ± 3.38 | 59.00 ± 2.25 | **58.83 ± 4.65** | 64.27 ± 1.92 |
| SAGE | 88.79 ± 0.23 | 82.42 ± 1.71 | **78.66 ± 2.02** | 82.56 ± 2.03 | 84.94 ± 1.29 | 79.49 ± 1.86 |
| Guard | 87.59 ± 0.20 | 67.83 ± 1.38 | **59.93 ± 0.87** | 62.31 ± 1.00 | 77.43 ± 1.26 | 66.77 ± 0.62 |
| GCN-LNi | 87.67 ± 0.20 | 74.09 ± 1.71 | **65.45 ± 1.29** | 69.39 ± 1.45 | 81.96 ± 1.19 | 70.18 ± 0.82 |

Table 22: WTGIA with average sparsity against more defense models on PubMed, BoW embedding.

| Model | Clean | SeqGIA | MetaGIA | TDGIA | ATDGIA | AGIA |
|---|---|---|---|---|---|---|
| GCN | 86.42 ± 0.24 | 43.87 ± 0.38 | 48.62 ± 0.26 | 48.33 ± 0.33 | **41.73 ± 0.37** | 45.24 ± 0.43 |
| GAT | 85.93 ± 0.16 | 45.62 ± 2.15 | 50.72 ± 1.77 | 50.50 ± 1.22 | **42.50 ± 2.26** | 47.19 ± 1.83 |
| SAGE | 87.00 ± 0.21 | 58.19 ± 2.01 | 60.67 ± 1.67 | 60.56 ± 1.69 | **57.88 ± 2.30** | 59.28 ± 1.87 |
| Guard | 86.54 ± 0.15 | **62.08 ± 0.60** | 64.64 ± 0.42 | 65.81 ± 0.40 | 62.54 ± 0.66 | 63.73 ± 0.65 |
| GCN-LNi | 86.36 ± 0.24 | 51.15 ± 1.32 | 54.28 ± 1.01 | 53.83 ± 1.06 | **50.05 ± 1.45** | 52.29 ± 1.35 |

results are still unsatisfactory. Although GTR and SBERT are both PLM-based embeddings, the adversarial text by ITGIA performs even worse than WTGIA when transferred to SBERT. We believe the transferability of text-level GIAs remains a significant challenge.

Table 23: Transferring ITGIA with default HAO weight to SBERT embedding on Cora.

| Model | Clean | SeqGIA | MetaGIA | TDGIA | ATDGIA | AGIA |
|-------|-------|--------|---------|-------|--------|------|
| BoW | 86.48 ± 0.41 | 84.85 ± 0.76 | **84.04 ± 0.78** | 85.56 ± 0.61 | 86.49 ± 0.50 | 84.90 ± 0.73 |
| GTR | 87.19 ± 0.62 | **65.67 ± 1.48** | 65.77 ± 1.15 | 71.49 ± 1.71 | 74.63 ± 2.48 | 68.59 ± 1.51 |
| SBERT | 89.12 ± 0.44 | 85.07 ± 0.85 | **82.97 ± 0.58** | 83.78 ± 0.76 | 86.69 ± 0.93 | 86.06 ± 0.65 |

Table 24: Transferring WTGIA with average sparsity to SBERT embedding on Cora.

| Model | Clean | SeqGIA | MetaGIA | TDGIA | ATDGIA | AGIA |
|-------|-------|--------|---------|-------|--------|------|
| BoW | 86.48 ± 0.41 | 84.85 ± 0.76 | **84.04 ± 0.78** | 85.56 ± 0.61 | 86.49 ± 0.50 | 84.90 ± 0.73 |
| GTR | 87.19 ± 0.62 | **65.67 ± 1.48** | 65.77 ± 1.15 | 71.49 ± 1.71 | 74.63 ± 2.48 | 68.59 ± 1.51 |
| SBERT | 89.12 ± 0.44 | 80.39 ± 1.43 | **78.01 ± 0.92** | 82.17 ± 0.89 | 85.85 ± 0.71 | 79.41 ± 1.16 |

## L  Full Experiment Results

### L.1  Figure 5 for WTGIA with Topic

The results of WTGIA variants with topic constrained in the prompt are shown in Figure 7.

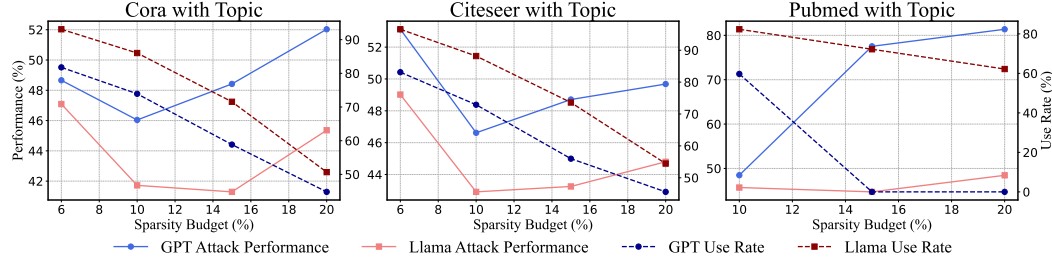

Figure 7: The performance of WTGIA(w topic) against GCN w.r.t sparsity budget. As the budget increases, the use rate keeps decreasing, and the attack performance increases and then decreases.

### L.2  Full Results for ITGIA

Full results for ITGIA, as shown in Figure 2 are displayed in Table 1, Table 26, Table 27, and Table 28.

Table 25: Performance of ITGIA without HAO. Raw texts are embedded by BoW, and GTR before being fed to GCN for evaluation.

| Dataset | Model | Clean | Random | SeqGIA | MetaGIA | TDGIA | ATDGIA | AGIA |
|---------|-------|-------|--------|--------|---------|-------|--------|------|
| Cora | BoW | 86.48 ± 0.41 | 84.89 ± 0.46 | 86.42 ± 0.56 | **84.86 ± 0.56** | 86.93 ± 0.65 | 86.10 ± 0.42 | 85.83 ± 0.54 |
| | GTR | 87.19 ± 0.62 | 82.73 ± 0.59 | 74.16 ± 1.76 | **71.35 ± 1.14** | 76.52 ± 1.45 | 76.73 ± 1.46 | 72.25 ± 1.32 |
| CiteSeer | BoW | 75.92 ± 0.55 | 75.50 ± 0.49 | 75.21 ± 0.67 | 75.48 ± 0.55 | 75.35 ± 0.52 | **74.67 ± 0.55** | 75.25 ± 0.39 |
| | GTR | 75.93 ± 0.41 | 74.79 ± 0.66 | 68.17 ± 0.94 | 69.39 ± 0.89 | 68.24 ± 1.30 | 69.72 ± 1.34 | **66.18 ± 1.19** |
| PubMed | BoW | 86.42 ± 0.24 | 85.38 ± 0.26 | 85.75 ± 0.21 | 85.41 ± 0.23 | 85.67 ± 0.18 | 86.36 ± 0.24 | **85.10 ± 0.30** |
| | GTR | 87.91 ± 0.26 | 82.36 ± 1.91 | 65.13 ± 1.67 | **58.96 ± 1.25** | 59.47 ± 1.08 | 69.81 ± 1.90 | 66.16 ± 0.97 |

### L.3  Full Results for Row-wise FGSM

Full results for Row-wise FGSM, as shown in Figure 3 are displayed in Table 29, Table 30, Table 31, Table 32, and Table 33.

Table 26: Performance of ITGIA with default HAO weight. Raw texts are embedded by BoW, and GTR before being fed to GCN for evaluation.

| Dataset | Model | Clean | Random | SeqGIA | MetaGIA | TDGIA | ATDGIA | AGIA |
|---|---|---|---|---|---|---|---|---|
| Cora | BoW | 86.48 ± 0.41 | 84.89 ± 0.46 | 85.23 ± 0.65 | **84.77 ± 0.55** | 85.56 ± 0.61 | 86.49 ± 0.50 | 86.05 ± 0.53 |
| | GTR | 87.19 ± 0.62 | 82.73 ± 0.59 | **65.67 ± 1.48** | 65.77 ± 1.15 | 71.49 ± 1.71 | 74.63 ± 2.48 | 68.59 ± 1.51 |
| CiteSeer | BoW | 75.92 ± 0.55 | 75.50 ± 0.49 | 75.02 ± 0.41 | 75.61 ± 0.59 | 74.89 ± 0.56 | 74.45 ± 0.68 | **73.77 ± 0.50** |
| | GTR | 75.93 ± 0.41 | 74.79 ± 0.66 | **64.79 ± 1.30** | 65.11 ± 1.01 | 67.43 ± 0.89 | 71.89 ± 0.50 | **64.79 ± 1.30** |
| PubMed | BoW | 86.42 ± 0.24 | 85.38 ± 0.26 | 85.76 ± 0.19 | 84.53 ± 0.42 | 85.35 ± 0.25 | 86.39 ± 0.16 | **84.41 ± 0.40** |
| | GTR | 87.91 ± 0.26 | 82.36 ± 1.91 | 66.40 ± 2.33 | **58.56 ± 1.22** | 60.26 ± 1.32 | 76.23 ± 2.08 | 65.77 ± 0.91 |

Table 27: Performance of ITGIA with 3x HAO weight. Raw texts are embedded by BoW, and GTR before being fed to GCN for evaluation.

| Dataset | Model | Clean | Random | SeqGIA | MetaGIA | TDGIA | ATDGIA | AGIA |
|---|---|---|---|---|---|---|---|---|
| Cora | BoW | 86.48 ± 0.41 | 84.89 ± 0.46 | 84.85 ± 0.76 | **84.04 ± 0.78** | 86.01 ± 0.76 | 85.58 ± 0.85 | 84.90 ± 0.73 |
| | GTR | 87.19 ± 0.62 | 82.73 ± 0.59 | **66.70 ± 0.94** | 67.83 ± 0.75 | 71.90 ± 1.92 | 72.93 ± 2.69 | 68.81 ± 1.39 |
| CiteSeer | BoW | 75.92 ± 0.55 | 75.50 ± 0.49 | **73.39 ± 0.69** | 75.17 ± 0.50 | 73.72 ± 0.57 | 74.37 ± 0.78 | 74.11 ± 0.60 |
| | GTR | 75.93 ± 0.41 | 74.79 ± 0.66 | 64.12 ± 1.14 | **60.77 ± 0.71** | 68.99 ± 1.24 | 71.75 ± 0.95 | 61.55 ± 1.39 |
| PubMed | BoW | 86.42 ± 0.24 | 85.38 ± 0.26 | 85.72 ± 0.25 | 85.17 ± 0.35 | **84.98 ± 0.24** | 86.33 ± 0.17 | 85.51 ± 0.20 |
| | GTR | 87.91 ± 0.26 | 82.36 ± 1.91 | 63.78 ± 2.41 | **59.74 ± 1.48** | 62.53 ± 1.92 | 78.31 ± 2.08 | 70.03 ± 1.56 |

Table 28: Performance of ITGIA with 5x HAO weight. Raw texts are embedded by BoW and GTR before being fed to GCN for evaluation.

| Dataset | Model | Clean | Random | SeqGIA | MetaGIA | TDGIA | ATDGIA | AGIA |
|---|---|---|---|---|---|---|---|---|
| Cora | BoW | 86.48 ± 0.41 | 84.89 ± 0.46 | 84.49 ± 0.86 | **83.06 ± 0.63** | 85.83 ± 1.02 | 85.62 ± 0.82 | 85.36 ± 0.85 |
| | GTR | 87.19 ± 0.62 | 82.73 ± 0.59 | 70.22 ± 0.89 | 77.28 ± 0.62 | 73.86 ± 1.93 | **65.19 ± 1.89** | 72.25 ± 1.28 |
| CiteSeer | BoW | 75.92 ± 0.55 | 75.50 ± 0.49 | **73.47 ± 0.74** | 75.58 ± 0.55 | 74.44 ± 0.55 | 74.57 ± 0.45 | 73.99 ± 0.91 |
| | GTR | 75.93 ± 0.41 | 74.79 ± 0.66 | 65.40 ± 1.01 | 72.97 ± 0.57 | 70.93 ± 1.14 | 72.26 ± 0.83 | **63.32 ± 1.31** |
| PubMed | BoW | 86.42 ± 0.24 | 85.38 ± 0.26 | 85.62 ± 0.19 | 84.91 ± 0.30 | **84.69 ± 0.28** | 86.50 ± 0.15 | 85.68 ± 0.26 |
| | GTR | 87.91 ± 0.26 | 82.36 ± 1.91 | 63.63 ± 2.42 | **60.12 ± 1.47** | 62.69 ± 2.08 | 77.93 ± 2.09 | 71.71 ± 1.78 |

Table 29: Performance of GCN and Guard under FGSM attack with budget being average sparsity of original dataset. Models are evaluated using BoW embeddings of raw texts.

| Dataset | Model | Clean | Random | SeqGIA | MetaGIA | TDGIA | ATDGIA | AGIA |
|---|---|---|---|---|---|---|---|---|
| Cora | GCN | 86.48 ± 0.41 | 85.00 ± 0.55 | 48.09 ± 0.73 | 51.12 ± 0.76 | 51.70 ± 1.08 | **34.11 ± 0.96** | 47.37 ± 0.68 |
| | Guard | 84.74 ± 0.35 | 83.78 ± 0.49 | 81.20 ± 0.43 | 81.41 ± 0.26 | 82.83 ± 0.40 | **79.16 ± 0.53** | 81.30 ± 0.37 |
| CiteSeer | GCN | 75.92 ± 0.55 | 74.73 ± 0.60 | 47.35 ± 0.62 | 51.20 ± 0.91 | 48.90 ± 0.93 | 49.04 ± 0.71 | **47.30 ± 0.79** |
| | Guard | 74.85 ± 0.49 | 74.26 ± 0.34 | 72.69 ± 0.44 | 72.48 ± 0.42 | 72.31 ± 0.38 | **71.69 ± 0.43** | 73.31 ± 0.42 |
| PubMed | GCN | 86.42 ± 0.24 | 83.72 ± 0.36 | 43.44 ± 0.25 | 48.11 ± 0.22 | 47.80 ± 0.36 | **41.30 ± 0.19** | 44.71 ± 0.38 |
| | Guard | 86.51 ± 0.17 | 84.73 ± 0.33 | 81.31 ± 0.39 | 82.32 ± 0.41 | 81.94 ± 0.33 | **80.89 ± 0.47** | 82.26 ± 0.36 |

Table 30: Performance of GCN and Guard under FGSM attack with a budget of 0.10 sparsity in the attacked embeddings. Models are evaluated using BoW embeddings of raw texts.

| Dataset | Model | Clean | Random | SeqGIA | MetaGIA | TDGIA | ATDGIA | AGIA |
|---|---|---|---|---|---|---|---|---|
| Cora | GCN | 86.48 ± 0.41 | 85.07 ± 0.56 | 43.75 ± 0.31 | 44.54 ± 0.39 | 43.70 ± 0.92 | **27.26 ± 0.71** | 42.23 ± 0.44 |
| | Guard | 84.74 ± 0.35 | 83.58 ± 0.39 | 77.07 ± 0.54 | 76.86 ± 0.52 | 79.15 ± 0.61 | **76.41 ± 0.70** | 77.81 ± 0.43 |
| CiteSeer | GCN | 75.92 ± 0.55 | 74.33 ± 0.62 | 42.68 ± 0.53 | 45.59 ± 0.68 | 41.06 ± 1.23 | 43.33 ± 0.69 | **39.02 ± 0.43** |
| | Guard | 74.85 ± 0.49 | 73.92 ± 0.67 | 70.41 ± 0.46 | 69.73 ± 0.63 | 69.33 ± 0.68 | **69.19 ± 0.41** | 69.99 ± 0.62 |

Table 31: Performance of GCN and Guard under FGSM attack with a budget of 0.15 sparsity in the attacked embeddings. Models are evaluated using BoW embeddings of raw texts.

| Dataset | Model | Clean | Random | SeqGIA | MetaGIA | TDGIA | ATDGIA | AGIA |
|---|---|---|---|---|---|---|---|---|
| Cora | GCN | 86.48 ± 0.41 | 84.36 ± 0.63 | 41.37 ± 0.32 | 41.98 ± 0.49 | 40.80 ± 0.38 | **23.75 ± 0.55** | 39.25 ± 0.26 |
| | Guard | 84.74 ± 0.35 | 83.06 ± 0.59 | 74.88 ± 0.57 | 72.74 ± 0.47 | 75.77 ± 0.38 | **71.06 ± 0.48** | 72.89 ± 0.28 |
| CiteSeer | GCN | 75.92 ± 0.55 | 74.57 ± 0.72 | 44.54 ± 0.41 | 42.68 ± 0.40 | **36.72 ± 0.97** | 37.42 ± 0.60 | 42.96 ± 0.26 |
| | Guard | 74.85 ± 0.49 | 73.59 ± 0.55 | 66.72 ± 0.45 | 66.65 ± 0.44 | 65.66 ± 0.47 | **64.15 ± 0.53** | 66.35 ± 0.31 |
| PubMed | GCN | 86.42 ± 0.24 | 82.53 ± 0.48 | 42.07 ± 0.22 | 46.41 ± 0.38 | 46.52 ± 0.45 | **40.61 ± 0.10** | 44.25 ± 0.34 |
| | Guard | 86.51 ± 0.17 | 83.47 ± 0.40 | 71.29 ± 0.86 | 72.15 ± 0.82 | 72.50 ± 0.55 | **70.09 ± 0.59** | 71.65 ± 0.84 |

Table 32: Performance of GCN and Guard under FGSM attack with a budget of 0.20 sparsity in the attacked embeddings. Models are evaluated using BoW embeddings of raw texts.

| Dataset | Model | Clean | Random | SeqGIA | MetaGIA | TDGIA | ATDGIA | AGIA |
|---|---|---|---|---|---|---|---|---|
| Cora | GCN | 86.48 ± 0.41 | 84.15 ± 0.70 | 39.80 ± 0.10 | 40.63 ± 0.44 | 40.23 ± 0.51 | **21.21 ± 0.26** | 41.15 ± 0.23 |
| | Guard | 84.74 ± 0.35 | 82.21 ± 0.36 | 71.63 ± 0.37 | 70.85 ± 0.38 | 72.65 ± 0.38 | **64.62 ± 0.66** | 70.90 ± 0.55 |
| CiteSeer | GCN | 75.92 ± 0.55 | 74.07 ± 0.72 | 39.69 ± 0.29 | 43.21 ± 0.43 | **33.88 ± 0.71** | 37.80 ± 0.32 | 41.57 ± 0.22 |
| | Guard | 74.85 ± 0.49 | 72.43 ± 0.69 | 60.62 ± 0.61 | 62.39 ± 0.56 | 60.29 ± 0.59 | **60.21 ± 0.38** | 64.42 ± 0.39 |
| PubMed | GCN | 86.42 ± 0.24 | 82.02 ± 0.45 | 41.97 ± 0.25 | 45.76 ± 0.41 | 45.86 ± 0.51 | **40.36 ± 0.10** | 43.71 ± 0.39 |
| | Guard | 86.51 ± 0.17 | 82.96 ± 0.96 | 62.26 ± 0.94 | 63.43 ± 0.80 | 64.21 ± 0.71 | **59.98 ± 0.76** | 63.90 ± 0.86 |

Table 33: Performance of GCN and Guard under FGSM attack with a budget of 0.25 sparsity in the attacked embeddings. Models are evaluated using BoW embeddings of raw texts.

| Dataset | Model | Clean | Random | SeqGIA | MetaGIA | TDGIA | ATDGIA | AGIA |
|---|---|---|---|---|---|---|---|---|
| Cora | GCN | 86.48 ± 0.41 | 84.21 ± 0.75 | 38.96 ± 0.16 | 39.88 ± 0.39 | 39.36 ± 0.41 | **21.89 ± 0.29** | 40.14 ± 0.13 |
| | Guard | 84.74 ± 0.35 | 81.28 ± 0.83 | 68.56 ± 0.51 | 65.72 ± 0.47 | 69.19 ± 0.52 | **57.58 ± 0.91** | 66.28 ± 0.44 |
| CiteSeer | GCN | 75.92 ± 0.55 | 74.15 ± 0.96 | 38.13 ± 0.47 | 40.59 ± 0.39 | **33.44 ± 0.86** | 35.57 ± 0.23 | 40.08 ± 0.28 |
| | Guard | 74.85 ± 0.49 | 71.61 ± 0.73 | 54.50 ± 0.46 | 55.91 ± 0.58 | **52.91 ± 0.51** | 55.10 ± 0.54 | 56.70 ± 0.44 |
| PubMed | GCN | 86.42 ± 0.24 | 81.76 ± 0.57 | 41.80 ± 0.31 | 45.05 ± 0.45 | 45.51 ± 0.60 | **40.08 ± 0.09** | 43.21 ± 0.46 |
| | Guard | 86.51 ± 0.17 | 83.01 ± 1.21 | 55.48 ± 0.74 | 56.47 ± 0.79 | 58.84 ± 0.67 | **53.62 ± 0.55** | 55.85 ± 0.92 |

## L.4 Full Results for WTGIA

Full results for WTGIA, as shown in Figure 5 are displayed in the below tables. For average sparsity budget: Table 34, Table 35, Table 36.

For sparsity budget 0.10: Table 37, Table 38.

For sparsity budget 0.15: Table 39, Table 40. Table 41.

For sparsity budget 0.20: Table 42, Table 43. Table 44.

Table 34: Performance of WTGIA on Cora. The sparsity budget is set as the average sparsity of the original dataset. The injected raw texts are embedded with BoW and GTR, and then evaluated using a GCN. The last column reports the average result of GIAs in the last five columns. The lowest(strongest) result in the column is bold.

| Generation Type | Embedding | Clean | Random | SeqGIA | MetaGIA | TDGIA | ATDGIA | AGIA | Avg Last Five |
|---|---|---|---|---|---|---|---|---|---|
| Embedding | BoW | 86.48 ± 0.41 | 85.00 ± 0.55 | 48.09 ± 0.73 | 51.12 ± 0.76 | 51.70 ± 1.08 | **34.11 ± 0.96** | 47.37 ± 0.68 | 46.48 ± 0.84 |
| GPT | BoW | 86.48 ± 0.41 | 84.94 ± 0.62 | 49.02 ± 0.68 | 52.14 ± 0.79 | 52.90 ± 1.12 | **36.02 ± 1.38** | 48.40 ± 0.69 | 47.70 ± 0.93 |
| | GTR | 87.19 ± 0.62 | 82.97 ± 0.66 | 78.04 ± 1.49 | **75.95 ± 1.06** | 79.00 ± 1.30 | 83.68 ± 1.06 | 77.25 ± 1.50 | 78.78 ± 1.28 |
| GPT-Topic | BoW | 86.48 ± 0.41 | 85.09 ± 0.55 | 49.58 ± 1.00 | 52.06 ± 0.86 | 53.75 ± 1.19 | **38.96 ± 1.94** | 48.95 ± 0.93 | 48.66 ± 1.18 |
| | GTR | 87.19 ± 0.62 | 83.07 ± 0.77 | 77.59 ± 1.62 | **76.41 ± 0.97** | 78.85 ± 1.49 | 83.57 ± 1.01 | 76.89 ± 1.54 | 78.66 ± 1.33 |
| Llama | BoW | 86.48 ± 0.41 | 84.99 ± 0.56 | 48.32 ± 0.74 | 51.58 ± 0.78 | 52.49 ± 1.32 | **35.33 ± 1.29** | 47.81 ± 0.78 | 47.11 ± 0.98 |
| | GTR | 87.19 ± 0.62 | 82.85 ± 0.78 | 78.15 ± 1.70 | **76.88 ± 0.96** | 79.27 ± 1.24 | 83.77 ± 1.11 | 77.95 ± 1.51 | 79.20 ± 1.30 |
| Llama-Topic | BoW | 86.48 ± 0.41 | 84.90 ± 0.62 | 48.57 ± 0.82 | 51.58 ± 0.73 | 52.30 ± 1.26 | **35.32 ± 1.13** | 47.69 ± 0.76 | **47.09 ± 0.94** |
| | GTR | 87.19 ± 0.62 | 82.84 ± 0.79 | 77.80 ± 1.74 | **76.42 ± 1.10** | 78.96 ± 1.26 | 83.52 ± 1.02 | 77.69 ± 1.46 | 78.88 ± 1.32 |

Table 35: Performance of WTGIA on CiteSeer. The sparsity budget is set as the average sparsity of the original dataset. The injected raw texts are embedded with BoW and GTR, and then evaluated using a GCN. The last column reports the average result of GIAs in the last five columns. The lowest(strongest) result in the column is bold.

| Generation Type | Embedding | Clean | Random | SeqGIA | MetaGIA | TDGIA | ATDGIA | AGIA | Avg Last Five |
|---|---|---|---|---|---|---|---|---|---|
| Embedding | BoW | 75.92 ± 0.55 | 74.73 ± 0.60 | 47.35 ± 0.62 | 51.20 ± 0.91 | 48.90 ± 0.93 | 49.04 ± 0.71 | **47.30 ± 0.79** | 48.76 ± 0.79 |
| GPT | BoW | 75.92 ± 0.55 | 75.02 ± 0.78 | 51.23 ± 0.68 | 51.97 ± 1.02 | 53.70 ± 1.42 | 55.81 ± 0.96 | **48.63 ± 0.84** | 52.27 ± 0.98 |
| | GTR | 75.93 ± 0.41 | 75.03 ± 0.56 | 68.50 ± 1.23 | **65.04 ± 0.85** | 69.43 ± 0.69 | 73.53 ± 0.55 | 71.24 ± 0.89 | 69.55 ± 0.84 |
| GPT-Topic | BoW | 75.92 ± 0.55 | 74.73 ± 0.72 | 53.17 ± 0.90 | 52.74 ± 1.06 | 54.48 ± 1.19 | 56.15 ± 0.79 | **49.06 ± 0.61** | 53.12 ± 0.91 |
| | GTR | 75.93 ± 0.41 | 74.72 ± 0.75 | 69.71 ± 1.23 | **64.17 ± 0.86** | 69.45 ± 0.58 | 73.26 ± 0.90 | 70.66 ± 0.73 | 69.45 ± 0.86 |
| Llama | BoW | 75.92 ± 0.55 | 74.58 ± 0.57 | **47.52 ± 0.65** | 51.25 ± 0.95 | 49.17 ± 0.94 | 49.50 ± 0.74 | 47.63 ± 0.81 | **49.01 ± 0.82** |
| | GTR | 75.93 ± 0.41 | 75.14 ± 0.58 | **65.21 ± 0.98** | 66.07 ± 0.77 | 70.15 ± 0.76 | 72.32 ± 0.71 | 69.82 ± 0.94 | 68.71 ± 0.83 |
| Llama-Topic | BoW | 75.92 ± 0.55 | 74.65 ± 0.64 | **47.60 ± 0.66** | 51.34 ± 0.98 | 49.09 ± 0.88 | 49.41 ± 0.79 | 47.66 ± 0.91 | 49.02 ± 0.84 |
| | GTR | 75.93 ± 0.41 | 75.51 ± 0.48 | **65.18 ± 1.10** | 65.84 ± 0.72 | 70.33 ± 0.70 | 72.72 ± 0.86 | 69.71 ± 0.83 | 68.76 ± 0.84 |

Table 36: Performance of WTGIA on PubMed. The sparsity budget is set as the average sparsity of the original dataset. The injected raw texts are embedded with BoW and GTR and then evaluated using a GCN. The last column reports the average result of GIAs in the last five columns. The lowest(strongest) result in the column is bold.

| Generation Type | Embedding | Clean | Random | SeqGIA | MetaGIA | TDGIA | ATDGIA | AGIA | Avg Last Five |
|---|---|---|---|---|---|---|---|---|---|
| Embedding | BoW | 86.42 ± 0.24 | 83.72 ± 0.36 | 43.44 ± 0.25 | 48.11 ± 0.22 | 47.80 ± 0.36 | **41.30 ± 0.19** | 44.71 ± 0.38 | 45.07 ± 0.28 |
| GPT | BoW | 86.42 ± 0.24 | 83.49 ± 0.33 | 44.86 ± 0.61 | 49.52 ± 0.33 | 49.22 ± 0.40 | **43.46 ± 0.86** | 46.29 ± 0.59 | 46.67 ± 0.56 |
| | GTR | 87.91 ± 0.26 | 83.53 ± 1.65 | 67.64 ± 2.43 | 67.23 ± 1.95 | **63.66 ± 2.11** | 68.59 ± 3.00 | 67.28 ± 2.26 | 66.88 ± 2.35 |
| GPT-Topic | BoW | 86.42 ± 0.24 | 82.65 ± 0.29 | 46.75 ± 1.00 | 50.71 ± 0.37 | 50.38 ± 0.48 | **46.03 ± 1.21** | 48.54 ± 0.85 | 48.48 ± 0.78 |
| | GTR | 87.91 ± 0.26 | 82.02 ± 1.48 | 64.99 ± 2.50 | 64.01 ± 1.67 | **61.23 ± 1.72** | 66.61 ± 2.96 | 64.65 ± 2.37 | 64.30 ± 2.24 |
| Llama | BoW | 86.42 ± 0.24 | 83.15 ± 0.38 | 43.87 ± 0.38 | 48.62 ± 0.26 | 48.33 ± 0.33 | **41.73 ± 0.37** | 45.24 ± 0.43 | **45.56 ± 0.35** |
| | GTR | 87.91 ± 0.26 | 82.82 ± 2.03 | 68.13 ± 2.34 | 65.92 ± 1.76 | **64.08 ± 2.05** | 69.38 ± 2.78 | 67.06 ± 2.19 | 66.91 ± 2.22 |
| Llama-Topic | BoW | 86.42 ± 0.24 | 83.01 ± 0.38 | 44.67 ± 0.59 | 48.54 ± 0.25 | 48.32 ± 0.38 | **41.76 ± 0.39** | 45.20 ± 0.44 | 45.70 ± 0.41 |
| | GTR | 87.91 ± 0.26 | 82.11 ± 2.10 | 66.66 ± 2.42 | 66.58 ± 1.76 | **64.78 ± 2.12** | 70.16 ± 2.82 | 67.84 ± 2.12 | 67.20 ± 2.25 |

Table 37: Performance of WTGIA on Cora. The sparsity budget of the embedding is set as 0.10. The injected raw texts are embedded with BoW and GTR, and then evaluated using a GCN. The last column reports the average result of GIAs in the last five columns. The lowest(strongest) result in the column is bold.

| Generation Type | Embedding | Clean | Random | SeqGIA | MetaGIA | TDGIA | ATDGIA | AGIA | Avg Last Five |
|---|---|---|---|---|---|---|---|---|---|
| Embedding | BoW | 86.48 ± 0.41 | 85.07 ± 0.56 | 43.75 ± 0.31 | 44.54 ± 0.39 | 43.70 ± 0.92 | **27.26 ± 0.71** | 42.23 ± 0.44 | 40.30 ± 0.55 |
| GPT | BoW | 86.48 ± 0.41 | 85.36 ± 0.49 | 45.04 ± 0.48 | 46.27 ± 0.70 | 47.63 ± 1.36 | **33.49 ± 1.61** | 44.00 ± 0.55 | 43.29 ± 0.94 |
| | GTR | 87.19 ± 0.62 | 82.95 ± 0.86 | 76.11 ± 1.48 | 76.04 ± 1.18 | 79.54 ± 1.24 | 81.79 ± 1.22 | **74.75 ± 1.64** | 77.65 ± 1.35 |
| GPT-Topic | BoW | 86.48 ± 0.41 | 86.15 ± 0.48 | 46.52 ± 0.65 | 47.44 ± 0.94 | 51.57 ± 1.56 | **39.43 ± 2.19** | 45.23 ± 0.80 | 46.04 ± 1.23 |
| | GTR | 87.19 ± 0.62 | 82.85 ± 0.84 | 75.79 ± 1.38 | 75.58 ± 1.13 | 79.47 ± 1.12 | 82.88 ± 1.24 | **74.62 ± 1.46** | 77.67 ± 1.27 |
| Llama | BoW | 86.48 ± 0.41 | 85.22 ± 0.61 | 44.33 ± 0.44 | 45.52 ± 0.65 | 45.56 ± 1.00 | **29.44 ± 1.05** | 43.06 ± 0.54 | **41.58 ± 0.74** |
| | GTR | 87.19 ± 0.62 | 83.11 ± 0.77 | 76.96 ± 1.48 | 76.74 ± 1.40 | 80.23 ± 1.29 | 82.23 ± 1.09 | **75.43 ± 1.53** | 78.32 ± 1.36 |
| Llama-Topic | BoW | 86.48 ± 0.41 | 85.39 ± 0.57 | 44.43 ± 0.45 | 45.64 ± 0.64 | 46.17 ± 1.10 | **29.35 ± 1.05** | 43.01 ± 0.45 | 41.72 ± 0.74 |
| | GTR | 87.19 ± 0.62 | 83.43 ± 0.99 | 76.99 ± 1.34 | 75.84 ± 1.13 | 80.22 ± 1.24 | 82.32 ± 1.18 | **74.73 ± 1.49** | 78.02 ± 1.28 |

Table 38: Performance of WTGIA on CiteSeer. The sparsity budget of the embedding is set as 0.10. The injected raw texts are embedded with BoW and GTR, and then evaluated using a GCN. The last column reports the average result of GIAs in the last five columns. The lowest(strongest) result in the column is bold.

| Generation Type | Embedding | Clean | Random | SeqGIA | MetaGIA | TDGIA | ATDGIA | AGIA | Avg Last Five |
|---|---|---|---|---|---|---|---|---|---|
| Embedding | BoW | 75.92 ± 0.55 | 74.33 ± 0.62 | 42.68 ± 0.53 | 45.59 ± 0.68 | 41.06 ± 1.23 | 43.33 ± 0.69 | **39.02 ± 0.43** | 42.34 ± 0.71 |
| GPT | BoW | 75.92 ± 0.55 | 74.58 ± 0.63 | 44.22 ± 0.73 | 46.43 ± 0.84 | 44.97 ± 1.14 | 47.49 ± 1.10 | **40.23 ± 0.65** | 44.67 ± 0.89 |
| | GTR | 75.93 ± 0.41 | 75.16 ± 0.69 | 71.18 ± 0.60 | **67.66 ± 0.94** | 70.70 ± 0.58 | 71.55 ± 0.97 | 68.17 ± 1.08 | 69.85 ± 0.83 |
| GPT-Topic | BoW | 75.92 ± 0.55 | 74.29 ± 0.74 | 45.64 ± 0.57 | 47.91 ± 0.97 | 46.84 ± 1.07 | 51.41 ± 1.09 | **41.31 ± 0.70** | 46.62 ± 0.88 |
| | GTR | 75.93 ± 0.41 | 75.23 ± 0.49 | 71.29 ± 0.57 | **68.43 ± 0.99** | 70.81 ± 0.66 | 71.86 ± 1.12 | 69.56 ± 1.24 | 70.39 ± 0.92 |
| Llama | BoW | 75.92 ± 0.55 | 74.31 ± 0.65 | 43.29 ± 0.57 | 46.10 ± 0.77 | 41.80 ± 1.24 | 44.10 ± 0.85 | **39.67 ± 0.45** | 42.99 ± 0.78 |
| | GTR | 75.93 ± 0.41 | 75.35 ± 0.54 | 70.51 ± 0.74 | **68.17 ± 1.01** | 70.77 ± 0.76 | 71.47 ± 1.09 | 68.89 ± 1.35 | 69.96 ± 0.99 |
| Llama-Topic | BoW | 75.92 ± 0.55 | 74.27 ± 0.65 | 43.42 ± 0.57 | 45.95 ± 0.78 | 41.84 ± 1.20 | 43.91 ± 0.72 | **39.42 ± 0.46** | 42.91 ± 0.75 |
| | GTR | 75.93 ± 0.41 | 75.41 ± 0.29 | 71.04 ± 0.82 | **69.34 ± 0.77** | 70.55 ± 0.61 | 71.95 ± 1.14 | 70.17 ± 1.42 | 70.61 ± 0.95 |

Table 39: Performance of WTGIA on Cora. The sparsity budget of the embedding is set as 0.15. The injected raw texts are embedded with BoW and GTR, and then evaluated using a GCN. The last column reports the average result of GIAs in the last five columns. The lowest(strongest) result in the column is bold.

| Generation Type | Embedding | Clean | Random | SeqGIA | MetaGIA | TDGIA | ATDGIA | AGIA | Avg Last Five |
|---|---|---|---|---|---|---|---|---|---|
| Embedding | BoW | 86.48 ± 0.41 | 84.36 ± 0.63 | 41.37 ± 0.32 | 41.98 ± 0.49 | 40.80 ± 0.38 | **23.75 ± 0.55** | 39.25 ± 0.26 | 37.23 ± 0.40 |
| GPT | BoW | 86.48 ± 0.41 | 85.31 ± 0.67 | 44.35 ± 0.54 | 45.04 ± 0.78 | 46.31 ± 1.47 | **41.28 ± 3.10** | 45.59 ± 0.90 | 44.51 ± 1.16 |
| | GTR | 87.19 ± 0.62 | 82.80 ± 0.71 | 78.04 ± 1.28 | **77.19 ± 0.97** | 80.90 ± 1.26 | 81.72 ± 1.10 | 77.84 ± 1.06 | 79.14 ± 1.13 |
| GPT-Topic | BoW | 86.48 ± 0.41 | 85.56 ± 0.63 | **46.32 ± 0.87** | 47.14 ± 0.86 | 51.57 ± 1.57 | 48.38 ± 3.06 | 48.68 ± 0.94 | 48.42 ± 1.46 |
| | GTR | 87.19 ± 0.62 | 83.32 ± 0.73 | **77.54 ± 1.19** | 77.77 ± 1.07 | 80.86 ± 1.29 | 82.09 ± 0.93 | 78.11 ± 0.94 | 79.27 ± 1.08 |
| Llama | BoW | 86.48 ± 0.41 | 84.68 ± 0.64 | 43.58 ± 0.54 | 44.05 ± 0.56 | 44.25 ± 1.23 | **30.07 ± 1.44** | 42.53 ± 0.83 | **40.90 ± 0.92** |
| | GTR | 87.19 ± 0.62 | 82.75 ± 0.84 | 76.52 ± 1.34 | **75.84 ± 1.01** | 80.16 ± 1.17 | 82.09 ± 0.86 | 77.28 ± 1.19 | 78.38 ± 1.11 |
| Llama-Topic | BoW | 86.48 ± 0.41 | 84.89 ± 0.52 | 43.19 ± 0.54 | 43.78 ± 0.51 | 43.85 ± 0.98 | **32.69 ± 1.53** | 42.94 ± 0.77 | 41.29 ± 0.87 |
| | GTR | 87.19 ± 0.62 | 83.14 ± 0.79 | 76.64 ± 1.17 | **75.81 ± 1.10** | 80.00 ± 1.22 | 81.47 ± 0.95 | 77.41 ± 1.02 | 78.27 ± 1.09 |

Table 40: Performance of WTGIA on CiteSeer. The sparsity budget of the embedding is set as 0.15. The injected raw texts are embedded with BoW and GTR, and then evaluated using a GCN. The last column reports the average result of GIAs in the last five columns. The lowest(strongest) result in the column is bold.

| Generation Type | Embedding | Clean | Random | SeqGIA | MetaGIA | TDGIA | ATDGIA | AGIA | Avg Last Five |
|---|---|---|---|---|---|---|---|---|---|
| Embedding | BoW | 75.92 ± 0.55 | 74.57 ± 0.72 | 44.54 ± 0.41 | 42.68 ± 0.40 | **36.72 ± 0.97** | 37.42 ± 0.60 | 42.96 ± 0.26 | 41.06 ± 0.53 |
| GPT | BoW | 75.92 ± 0.55 | 74.86 ± 0.90 | 47.74 ± 0.61 | 45.53 ± 0.56 | **44.71 ± 1.30** | 44.93 ± 1.07 | 46.09 ± 0.40 | 45.80 ± 0.79 |
| | GTR | 75.93 ± 0.41 | 75.22 ± 0.73 | 70.86 ± 0.79 | 71.43 ± 0.73 | **70.75 ± 0.76** | 70.85 ± 1.24 | 72.33 ± 0.76 | 71.24 ± 0.86 |
| GPT-Topic | BoW | 75.92 ± 0.55 | 74.92 ± 0.59 | 49.66 ± 0.64 | **46.97 ± 0.52** | 48.43 ± 1.20 | 50.29 ± 1.18 | 48.21 ± 0.49 | 48.71 ± 0.81 |
| | GTR | 75.93 ± 0.41 | 75.20 ± 0.87 | **70.45 ± 0.71** | 71.70 ± 0.72 | 71.17 ± 0.59 | 71.96 ± 1.07 | 72.58 ± 0.71 | 71.57 ± 0.76 |
| Llama | BoW | 75.92 ± 0.55 | 74.89 ± 0.77 | 45.57 ± 0.49 | 43.73 ± 0.46 | **39.05 ± 0.92** | 40.26 ± 0.97 | 44.11 ± 0.42 | **42.54 ± 0.65** |
| | GTR | 75.93 ± 0.41 | 75.39 ± 0.61 | **70.42 ± 0.60** | 70.83 ± 0.76 | 70.94 ± 0.57 | 71.47 ± 1.08 | 72.43 ± 0.96 | 71.22 ± 0.79 |
| Llama-Topic | BoW | 75.92 ± 0.55 | 74.60 ± 0.74 | 46.56 ± 0.58 | 44.28 ± 0.49 | 40.82 ± 1.03 | **40.21 ± 0.78** | 44.40 ± 0.44 | 43.25 ± 0.66 |
| | GTR | 75.93 ± 0.41 | 75.15 ± 0.45 | **70.29 ± 0.77** | 70.78 ± 0.72 | 71.19 ± 0.67 | 71.51 ± 1.14 | 71.75 ± 0.58 | 71.10 ± 0.78 |

Table 41: Performance of WTGIA on PubMed. The sparsity budget of the embedding is set as 0.15. The injected raw texts are embedded with BoW and GTR, and then evaluated using a GCN. The last column reports the average result of GIAs in the last five columns. The lowest(strongest) result in the column is bold. Note that GPT-Topic fails to give a meaningful response.

| Generation Type | Embedding | Clean | Random | SeqGIA | MetaGIA | TDGIA | ATDGIA | AGIA | Avg Last Five |
|---|---|---|---|---|---|---|---|---|---|
| Embedding | BoW | 86.42 ± 0.24 | 82.53 ± 0.48 | 42.07 ± 0.22 | 46.41 ± 0.38 | 46.52 ± 0.45 | **40.61 ± 0.10** | 44.25 ± 0.34 | 43.97 ± 0.30 |
| GPT | BoW | 86.42 ± 0.24 | 82.69 ± 0.42 | 44.19 ± 0.55 | 48.65 ± 0.25 | 49.04 ± 0.39 | **43.61 ± 0.70** | 47.11 ± 0.58 | 46.52 ± 0.49 |
| | GTR | 87.91 ± 0.26 | 83.06 ± 1.90 | 66.37 ± 2.30 | 65.05 ± 1.77 | **63.19 ± 2.07** | 67.83 ± 2.81 | 66.68 ± 1.95 | 65.82 ± 2.18 |
| Llama | BoW | 86.42 ± 0.24 | 82.39 ± 0.51 | 42.62 ± 0.31 | 47.24 ± 0.26 | 47.66 ± 0.39 | **41.02 ± 0.28** | 45.15 ± 0.42 | 44.74 ± 0.33 |
| | GTR | 87.91 ± 0.26 | 82.35 ± 2.05 | 66.19 ± 2.07 | 64.49 ± 1.70 | **62.78 ± 1.84** | 68.12 ± 2.60 | 66.66 ± 1.94 | 65.65 ± 2.03 |
| Llama-Topic | BoW | 86.42 ± 0.24 | 82.28 ± 0.32 | 42.58 ± 0.30 | 47.18 ± 0.26 | 47.71 ± 0.38 | **40.99 ± 0.32** | 45.13 ± 0.45 | **44.72 ± 0.34** |
| | GTR | 87.91 ± 0.26 | 81.72 ± 2.02 | 66.66 ± 2.16 | 65.21 ± 1.77 | **63.65 ± 1.94** | 68.30 ± 2.65 | 67.61 ± 1.95 | 66.29 ± 2.09 |

Table 42: Performance of WTGIA on Cora. The sparsity budget of the embedding is set as 0.20. The injected raw texts are embedded with BoW and GTR, and then evaluated using a GCN. The last column reports the average result of GIAs in the last five columns. The lowest(strongest) result in the column is bold.

| Generation Type | Embedding | Clean | Random | SeqGIA | MetaGIA | TDGIA | ATDGIA | AGIA | Avg Last Five |
|---|---|---|---|---|---|---|---|---|---|
| Embedding | BoW | 86.48 ± 0.41 | 84.15 ± 0.70 | 39.80 ± 0.10 | 40.63 ± 0.44 | 40.23 ± 0.51 | **21.21 ± 0.26** | 41.15 ± 0.23 | 36.68 ± 0.31 |
| GPT | BoW | 86.48 ± 0.41 | 85.16 ± 0.53 | **44.41 ± 0.87** | 46.46 ± 0.73 | 50.58 ± 1.36 | 47.83 ± 2.69 | 46.22 ± 0.85 | 47.10 ± 1.30 |
| | GTR | 87.19 ± 0.62 | 82.80 ± 0.83 | 79.72 ± 1.22 | 79.27 ± 1.01 | 81.53 ± 1.24 | 81.46 ± 1.27 | **78.36 ± 1.29** | 80.07 ± 1.21 |
| GPT-Topic | BoW | 86.48 ± 0.41 | 85.37 ± 0.63 | **48.80 ± 1.68** | 49.00 ± 1.20 | 56.96 ± 1.65 | 55.16 ± 2.25 | 50.23 ± 1.38 | 52.03 ± 1.63 |
| | GTR | 87.19 ± 0.62 | 82.37 ± 0.93 | 79.68 ± 1.47 | 79.12 ± 0.95 | 81.43 ± 1.24 | 82.27 ± 1.05 | **78.27 ± 1.06** | 80.15 ± 1.15 |
| Llama | BoW | 86.48 ± 0.41 | 84.70 ± 0.65 | **42.84 ± 0.70** | 45.14 ± 0.64 | 49.35 ± 1.32 | 47.49 ± 2.71 | 44.20 ± 0.58 | 45.80 ± 1.19 |
| | GTR | 87.19 ± 0.62 | 82.62 ± 0.64 | 77.93 ± 1.30 | 77.78 ± 1.12 | 80.84 ± 1.02 | 81.51 ± 0.99 | **76.74 ± 1.47** | 78.96 ± 1.18 |
| Llama-Topic | BoW | 86.48 ± 0.41 | 85.09 ± 0.49 | 43.83 ± 0.81 | 45.89 ± 0.77 | 48.99 ± 1.20 | **43.11 ± 2.48** | 44.96 ± 0.71 | **45.36 ± 1.19** |
| | GTR | 87.19 ± 0.62 | 82.79 ± 0.78 | 77.26 ± 1.28 | 77.58 ± 1.03 | 80.22 ± 1.00 | 81.81 ± 1.07 | **76.51 ± 1.28** | 78.68 ± 1.13 |

Table 43: Performance of WTGIA on CiteSeer. The sparsity budget of the embedding is set as 0.20. The injected raw texts are embedded with BoW and GTR, and then evaluated using a GCN. The last column reports the average result of GIAs in the last five columns. The lowest(strongest) result in the column is bold.

| Generation Type | Embedding | Clean | Random | SeqGIA | MetaGIA | TDGIA | ATDGIA | AGIA | Avg Last Five |
|---|---|---|---|---|---|---|---|---|---|
| Embedding | BoW | 75.92 ± 0.55 | 74.07 ± 0.72 | 39.69 ± 0.29 | 43.21 ± 0.43 | **33.88 ± 0.71** | 37.80 ± 0.32 | 41.57 ± 0.22 | 39.23 ± 0.39 |
| GPT | BoW | 75.92 ± 0.55 | 74.72 ± 0.83 | **45.07 ± 0.62** | 47.45 ± 0.58 | 45.58 ± 0.94 | 49.16 ± 0.85 | 46.25 ± 0.62 | 46.70 ± 0.72 |
| | GTR | 75.93 ± 0.41 | 74.79 ± 0.76 | **70.04 ± 0.56** | 70.55 ± 0.74 | 72.46 ± 0.67 | 71.96 ± 1.04 | 72.42 ± 0.81 | 71.49 ± 0.76 |
| GPT-Topic | BoW | 75.92 ± 0.55 | 74.65 ± 0.73 | **47.41 ± 0.74** | 49.06 ± 0.52 | 49.64 ± 1.01 | 54.43 ± 1.17 | 47.85 ± 1.01 | 49.68 ± 0.89 |
| | GTR | 75.93 ± 0.41 | 74.81 ± 0.71 | **71.14 ± 0.78** | 71.53 ± 0.73 | 73.34 ± 0.57 | 72.44 ± 1.17 | 72.28 ± 0.52 | 72.15 ± 0.75 |
| Llama | BoW | 75.92 ± 0.55 | 74.54 ± 0.81 | 43.00 ± 0.59 | 45.28 ± 0.54 | **41.00 ± 1.27** | 44.12 ± 0.74 | 43.84 ± 0.32 | **43.45 ± 0.69** |
| | GTR | 75.93 ± 0.41 | 75.01 ± 0.63 | **70.16 ± 0.71** | 70.79 ± 0.75 | 72.64 ± 0.61 | 72.45 ± 0.94 | 72.14 ± 0.62 | 71.64 ± 0.73 |
| Llama-Topic | BoW | 75.92 ± 0.55 | 74.81 ± 0.89 | 44.32 ± 0.80 | 46.20 ± 0.50 | **42.95 ± 1.14** | 46.14 ± 0.79 | 44.41 ± 0.43 | 44.80 ± 0.73 |
| | GTR | 75.93 ± 0.41 | 75.26 ± 0.48 | **69.80 ± 0.74** | 70.75 ± 0.85 | 72.59 ± 0.82 | 71.95 ± 1.30 | 72.17 ± 0.40 | 71.45 ± 0.82 |

Table 44: Performance of WTGIA on PubMed. The sparsity budget of the embedding is set as 0.20. The injected raw texts are embedded with BoW and GTR, and then evaluated using a GCN. The last column reports the average result of GIAs in the last five columns. The lowest(strongest) result in the column is bold. Note that GPT-Topic fails to give a meaningful response.

| Generation Type | Embedding | Clean | Random | SeqGIA | MetaGIA | TDGIA | ATDGIA | AGIA | Avg Last Five |
|---|---|---|---|---|---|---|---|---|---|
| Embedding | BoW | 86.42 ± 0.24 | 82.02 ± 0.45 | 41.97 ± 0.25 | 45.76 ± 0.41 | 45.86 ± 0.51 | **40.36 ± 0.10** | 43.71 ± 0.39 | 43.53 ± 0.33 |
| GPT | BoW | 86.42 ± 0.24 | 83.53 ± 0.29 | **47.89 ± 0.81** | 49.82 ± 0.35 | 51.49 ± 0.52 | 48.06 ± 1.10 | 48.73 ± 0.52 | 49.20 ± 0.66 |
| | GTR | 87.91 ± 0.26 | 83.74 ± 1.27 | 67.51 ± 2.21 | 66.43 ± 1.85 | **65.86 ± 2.33** | 69.88 ± 2.49 | 66.56 ± 2.10 | 67.25 ± 2.20 |
| Llama | BoW | 86.42 ± 0.24 | 81.92 ± 0.39 | 46.61 ± 0.74 | 49.09 ± 0.34 | 50.23 ± 0.42 | **46.11 ± 1.15** | 48.53 ± 0.60 | **48.11 ± 0.65** |
| | GTR | 87.91 ± 0.26 | 82.19 ± 1.94 | 64.28 ± 2.02 | 63.06 ± 1.75 | **61.86 ± 1.69** | 66.88 ± 2.45 | 64.64 ± 1.78 | 64.14 ± 1.94 |
| Llama-Topic | BoW | 86.42 ± 0.24 | 81.52 ± 0.40 | **46.80 ± 0.77** | 49.22 ± 0.31 | 50.67 ± 0.41 | 46.88 ± 1.26 | 48.69 ± 0.58 | 48.45 ± 0.67 |
| | GTR | 87.91 ± 0.26 | 79.53 ± 2.18 | 65.43 ± 2.12 | 63.04 ± 1.79 | **62.21 ± 1.87** | 68.25 ± 2.32 | 64.78 ± 1.90 | 64.74 ± 2.00 |

