# OpenReview forum: "Intruding with Words: Towards Understanding Graph Injection Attacks at the Text Level"
_NeurIPS.cc/2024/Conference — NeurIPS 2024 poster_

### Official Review · Reviewer_Da4q · 2024-07-09

**Soundness:** 4
**Presentation:** 3
**Contribution:** 4
**Rating:** 7
**Confidence:** 4

**Summary:**

The paper studies Graph Injection Attacks on text-attributed graphs. The study presents three attack designs: Vanilla Text-level GIA (VTGIA), Inversion-based Text-level GIA (ITGIA), and Word-frequency-based Text-level GIA (WTGIA). The key contributions include demonstrating the effectiveness of text level perturbation on Graph Injection Attack and the performance of using LLMs as a defender. WTGIA shows significant attack potential while maintaining understandability. It is a timely study on the vulnerabilities of text attributes in graphs.

**Strengths:**

1. Good investigation on the role of text manipulation in graph injection attacks. The text attributes in graphs are getting increasingly important. Demonstrating their vulnerabilities to GIA and the role of LLM can play as a defender are important to applications using text-attributed graphs.

2. The proposed WTGIA method achieves good attack performance with understandable text perturbation.

**Weaknesses:**

1. The embedding-text-embedding process is not very clear to me. Additional explanation will be helpful. It is unclear how different embedding models may affect the performance of this process.

2. The term interpretability in this paper has different semantics to the commonly used one in explainable AI. It is confusing and creates unnecessary obstacles in reading. It is better replace with understandability.

**Questions:**

1. What is the computing complexity of WTGIA? How long does it take to generate an attack in average in your experiment environment?

2. What is the impact of different text embedding models on the attack performance?

**Limitations:**

Limitations on transferability is discussed in the paper.

---

> ### Author Rebuttal · Authors · 2024-08-06
>
> Thank you for your thoughtful feedback on our paper. We have addressed your concerns and provided clarifications below.
>
> ---
>
> **W1:** *More about the Embedding-Text-Embedding Process.*
>
> **Why Embedding-to-Text:** Traditional GIAs included only the text-to-embedding process, with the pipeline as follows: 1) Preprocess the original graph text to embeddings, denoted as A.
> 2) The attacker designs an attack on this embedding by injecting abnormal embeddings, denoted as B.
> 3) The defender evaluates the same embeddings (A and B).
> There are several unreasonable aspects in this process, including: embedding B may not be reasonable at the text level (ITGIA); in real scenarios, attackers cannot directly inject embeddings into the defender's system, making step 3 overly idealistic for the attacker; and the defender can choose a different process in step 1 compared to the attacker.
> These issues arise because the attacker did not consider the embedding-to-text process.
> Our paper focuses on supplementing this step, requiring the attacker to generate text at the text level, injecting the original text.
>
> **Why Text-to-Embedding:** The embedding-text-embedding process is ultimately to align with previous GNN evaluations, as the GNN ultimately uses embeddings as input. So we finally transform the generated text into embeddings for evaluation. An exception is the LLM-as-predictor in Section 5, which only includes the embedding-to-text process. In the future, we believe that studying graph attacks and defenses involving embedding-to-text will be a fascinating direction.
>
> We will provide a clearer illustration of the Embedding-Text-Embedding process in the revised versions.
>
> ---
>
> **W2:** *Use of the term Interpretability*
>
> Thank you very much for your clarification, which helps avoid potential misunderstandings we might have caused.
> We will consider clarifying the distinction between this term and interpretability in the field of explainability when it is first mentioned, emphasizing that it is more about understandability to humans.
>
> ---
>
> **Q1:** *Running Time*
>
> We have included the running time and discussion about complexity in the **author rebuttal**.
> In summary, scalability is not a major issue in current text-level GIAs.
>
> ---
>
> **Q2:** *Impact of Embedding Models*
>
> We added a new embedding model, SentenceBert, and related experiments in the **author rebuttal**.
> Overall, the impact of text-embedding includes:
> 1) **Impact on the Original GNN:** According to [1], different embeddings can cause performance differences in GNNs on clean datasets. For example, on the Reddit dataset, GTR can have more than a 5-point performance advantage over the original dataset, which is significant in some cases.
> 2) **Impact on Attacker Design:** Designing a better attack for continuous embeddings can be more challenging. WTGIA shows some transferability across both, but the results are still unsatisfactory. Although GTR and SBERT are both PLM-based embeddings, the adversarial text by ITGIA performs even worse than WTGIA when transferred to SBERT. We believe that exploring this aspect remains a challenge.
> 3) **Impact on Defender Design:** LayerNorm is effective in defending against continuous features because it can capture structural anomalies caused by attacks on continuous features, but it performs mediocrely with BoW methods. While EGNNGuard can be easily bypassed by HAO, it remains competitive when defending against WTGIA. Therefore, different defense methods may be suitable for different types of embeddings.
>
> To sum up, there are still many areas worth exploring in this regard, and we believe considering more combinations of graph and text methods will be an interesting research direction in the future.
>
> ---
>
> Thank you once again for your valuable feedback and suggestions. We believe that incorporating these insights will significantly enhance the quality and impact of our work. We welcome any further questions or discussions.
>
> ---
>
> **Reference**
>
> [1] Chen, et al. Exploring the potential of large language models (LLMs) in learning on graphs.

---

> > ### Comment · Reviewer_Da4q · 2024-08-12
> >
> > Thanks for the response and additional experimental results. These largely address my concerns.

---

> > > ### Author Response · Authors · 2024-08-12
> > >
> > > Thank you for your appreciation of our work.
> > > We will carefully consider your suggestions to further improve our paper.

---

### Official Review · Reviewer_nfHh · 2024-07-10

**Soundness:** 4
**Presentation:** 3
**Contribution:** 4
**Rating:** 7
**Confidence:** 4

**Summary:**

This paper investigates an interesting topic. The authors first discuss the practicality of Graph Injection Attacks (GIA), arguing that previous approaches, which merely inject harmful embeddings, are unrealistic. It introduces three text-level GIA methods: ITGIA, VTGIA, and WTGIA, and conducts a thorough examination of their attacking performance and interpretability. The findings reveal that WTGIA excels in both attacking effectiveness and interpretability. Additionally, the paper explores the defensive capabilities of Large Language Models (LLMs) and discovers that they can effectively evade the impact of injected nodes. This underscores the importance of enhancing the performance of text-level attackers.

**Strengths:**

1. This paper first addresses the impracticality of embedding-based GIAs and explores text-level GIAs, introducing three novel text-level GIA methods. Among these, WTGIA effectively balances attack performance and interpretability.

2. The authors have conducted comprehensive experiments that thoroughly address the research questions posed in the study.

3. The theorem discusses the conditions under which destructive texts can be identified, providing a clear theoretical foundation for the work.

4. This paper offers valuable insights into the new challenges associated with text-level GIAs, identifying emerging directions in this field.

**Weaknesses:**

1. The authors should provide references for the cited papers on TDGIA, ATDGIA, MetaGIA, and AGIA.

2. The authors should explain why the attack performance initially decreases and then increases as the sparsity budget increases, as shown in Figure 5.

3. The authors should clarify how the LLM optimizes the graph structure after generating the text for the injected nodes.

**Questions:**

1. For VTGIA, how does the LLM optimize the graph structure after generating the text for the injected nodes?

2. The attacking method of WTGIA is interesting. Could this idea inspire other areas of GNN?

3. In Section 4, could you clarify the embedding-text-embedding process? Specifically, in Section 3, under Analysis of Performance and Interpretability, it seems that after inversion, text is generated. What embedding results after this inversion?

4. Why do you introduce FGSM to generate binary embeddings? Is it not possible to directly transform the embeddings into binary embeddings?

5. In Figure 5, why does the attack performance initially decrease and then increase as the sparsity budget increases?

**Limitations:**

Some parts of Section 4.1 are hard to understand, I admit I didn't check the proof related to the theorem in the Appendix. I think the authors could add figures or improve their presentations to better illustrate WTGIA.

---

> ### Author Rebuttal · Authors · 2024-08-06
>
> Thank you for your insightful comments and suggestions. Please find our detailed responses below:
>
> ---
>
> **W1:** *Citations of TDGIA, ATDGIA, MetaGIA, and AGIA.*
>
> We have cited TDGIA in Section 2.
> The other methods (ATDGIA, MetaGIA, and AGIA) are introduced in [1].
> In our experiments, we followed [1]'s implementation using a sequential optimization pipeline, which is detailed in the Appendix.
> We will ensure that the citations and explanations are clarified in future versions of the paper.
>
> ---
>
> **W2, Q5:** *Why attack performance decreases and then increases as the sparsity budget increases.*
>
> This phenomenon is explained in lines 287-294 of the main text.
> LLMs prioritize maintaining coherence and fluency in their outputs.
> When the number of required words increases within a fixed text length, the Use Rate of these words decreases, leading to reduced accuracy in the embedding-text-embedding process.
> This results in a decline in attack performance when the sparsity budget exceeds a certain threshold.
> Appendix J provides a more detailed illustration, quantifying coherence using Perplexity.
>
> ---
>
> **W3, Q1:** *How graph structure is optimized.*
>
> The LLM is not involved in the optimization of graph structures.
> Appendix I (Algorithm 4) provides pseudocode demonstrating that we rely on embedding-level techniques for selecting graph structures.
> After text generation, we employ structure optimization techniques from [1] to generate the graph structure.
> We maintain a fixed feature set once generated and do not alter the pipeline.
> Using LLMs for graph structure optimization is an intriguing direction for future research, but current studies are limited, and scalability issues need to be addressed.
>
> ---
>
> **Q2:** *Transferring WTGIA to more fields.*
>
> The WTGIA concept transforms text generation tasks into word-to-text tasks, enabling LLMs to effectively complete generation tasks while maintaining downstream classification performance.
> Given the importance of classification in graph representation learning, we believe this approach simplifies the generation tasks for related fields.
>
> ---
>
> **Q3:** *Embedding-Text-Embedding process.*
>
> Our paper focuses on the embedding-to-text process, introducing three text-level GIAs. The text-to-embedding process is necessary because, except for the LLM-as-predictor (Section 5), GNNs require converting injected text into embeddings as input. We utilize embedding methods like BoW and GTR to facilitate GNN testing on downstream tasks. During this process, the ITGIA method experienced significant accuracy loss. This loss occurs because, even with the same embedding method, the inverted text’s embedding differs significantly from the original. Section 3 discusses the reasons in detail, emphasizing the challenges of the ill-defined interpretable region.
>
> ---
>
> **Q4:** *Use of FGSM*
>
> FGSM is a classic optimization method suitable for binary states (0 and 1).
> It uses flipping for perturbation, preserving physical meaning, such as 0-1 for using a word and 1-0 for not using a word.
> As a result, it facilitates the later embedding-to-text process in WTGIA.
>
> Converting continuous embeddings to binary is challenging and relies on text as a bridge.
> What's more, the accuracy of the embedding-to-text process is crucial, posing a challenge for continuous features.
>
> ---
>
> Thank you again for your valuable feedback.
>
> ---
>
> **Reference**
>
> [1] Chen, et al. Understanding and improving graph injection attack by promoting unnoticeability.

---

> > ### Comment · Reviewer_nfHh · 2024-08-12
> > **Response to Authors**
> >
> > Thanks for the detailed response. I am pleased to note that you have addressed all the concerns I raised. I appreciate your efforts in clarifying these points and I have raised the scores.

---

> > > ### Author Response · Authors · 2024-08-13
> > >
> > > Thank you for your appreciation of our work. We will carefully consider your suggestions to further improve our paper.

---

### Official Review · Reviewer_ZG7Z · 2024-07-12

**Soundness:** 4
**Presentation:** 3
**Contribution:** 4
**Rating:** 8
**Confidence:** 4

**Summary:**

The paper studies GIAs, particularly focusing on text-attributed graphs (TAGs). It introduces a method for GIAs by injecting textual content directly into the graph, as opposed to the traditional method of embedding-level attacks. The authors propose three new attack designs: Vanilla Text-level GIA (VTGIA), Inversion-based Text-level GIA (ITGIA), and Word-frequency-based Text-level GIA (WTGIA). They demonstrate that text interpretability is necessary for the effectiveness of these attacks. They also show that defenses can be enhanced using customized text embedding methods or large language model (LLM)–based predictors.

**Strengths:**

I found this to be a very original paper as it shifts from embedding-level GIAs to text-level GIAs. Three new attacks are created and are tested theoretically and empirically.  Most importantly, the GIAs created ensure that the attacked nodes have semantically useful information, making the attack more realistic, using LLMs.

**Weaknesses:**

The paper does not explore the robustness of the proposed GIAs against more sophisticated defense mechanisms beyond customized text embedding methods and LLM-based predictors.

Although the authors mention that LLM-based defenses enhance protection against text-level GIAs, the experiments primarily focus on a few specific methods and datasets. These are limited in scope.

The trade-offs between performance, unnoticeability, and interpretability, as discussed in Sections 4.1 and 4.2, could be examined more thoroughly. While the paper shows that increasing the sparsity budget enhances attack performance but reduces interpretability, further experiments across a wider variety of datasets could validate these findings more convincingly, especially with additional baselines. Including more traditional GIA methods or other recent advancements in graph attack techniques would also be useful.

The discussion on the practical implementation of these attacks in real-world applications could be expanded. The paper mainly provides experimental insights but lacks detailed exploration of the real-world scenarios you stake your paper around, and practical challenges like the scalability of the proposed methods to larger and more complex graphs, the computational costs associated with implementing these attacks, and potential detection mechanisms in real-world systems are not thoroughly addressed. Providing more actionable insights and guidelines for practitioners could significantly enhance the practical relevance of the research.

Lastly, while the authors discuss the potential for enhancing defenses using customized text embedding methods or LLM-based predictors, they do not explore the possibility of integrating multiple defense strategies. Combining different defense mechanisms could potentially create more robust protection schemes against text-level GIAs.

Typos: line 296 and line 211

**Questions:**

1. Can the authors provide more details on how the sparsity budget impacts the interpretability and effectiveness of the text-level GIAs across different datasets?
2. How would this work apply to dynamic graphs where nodes and edges evolve over time?
3. Are there any challenges in scaling the proposed GIAs to larger graphs or more complex datasets?

---

> ### Author Rebuttal · Authors · 2024-08-06
>
> Thank you for your detailed feedback on our paper. We appreciate your insights and have addressed your concerns below.
>
> ---
>
> **W1 and W5:** *Exploration of Robustness Against Defense Mechanisms*
>
> In the **author rebuttal**, we included classic methods like GAT and GraphSAGE, as well as the EGNNGuard and Layernorm (LN) methods, which have been proven highly effective in [1].
> We indeed found interesting results, such as LN’s effectiveness on text-level GIAs not being as strong as previously reported for embedding-level attacks.
> This is because embedding-level GIAs often exhibit abnormal norms, which LN exploits for defense.
> However, text-level GIAs derive embeddings from real text, avoiding structural anomalies and bypassing LN’s defenses. This also means that some traditional defense methods that were particularly effective may actually be limited to the embedding level.
> The EGNNGuard method is generally effective and performs well overall. As we mentioned in the main paper, attackers can add tricks like HAO to bypass EGNNGuard, but this often involves trade-offs.
>
> Regarding using LLM-based methods for defense, we believe the current LLM for Graph methods still face the following challenges: 1) There is still a lack of dedicated methods combining LLM and Graph for defense. 2) The LLM-as-Predictors [2] method still faces scalability issues, and directly using it on large graphs requires significant resources. 3) LLM-as-Enhancer [2] methods still rely on GNN as the backbone, making them not much different from using defensive GNNs in terms of defense.
> Considering these challenges, we used predictors as an initial exploration. We believe that further exploring the combination of LLM and GNN to enhance defense capabilities will be an interesting direction.
>
> ---
>
> **W2, W3, Q1:** *More Datasets, More GIAs, More about the trade-off*
>
> In terms of datasets, we selected the most common GIA test datasets in the original paper.
> In the **author rebuttal**, we added larger datasets, such as ogbn-arxiv, and the social network dataset Reddit.
> We demonstrated that WTGIA can be applied to larger graphs (over 160,000 nodes) and different domains.
> Considering that the complexity of VTGIA and ITGIA is lower than that of WTGIA (see running time), we believe this sufficiently demonstrates the versatility of the text-level GIA framework.
> In the future, we believe it is interesting to explore the differences in text-level GIA across various domains.
>
> For GIAs, we selected representative and competitive GIA models for evaluation.
> For instance, TDGIA is a top-ranked model on the GRB benchmark [3], while the SeqGIA family [1] effectively addresses unnoticeability with desirable efficiency.
> For embeddings, we included PGD, FGSM, and feature-smooth (TDGIA/ATDGIA), three representative learnable embeddings. Therefore, we believe the current GIAs are sufficient to support our conclusions.
> We are open to incorporating new GIAs as needed in the future.
>
> Regarding the trade-off, we have more discussion in Appendix J.
> For example, we found that larger models tend to reduce the Use Rate to ensure the fluency of the output, selecting words that more easily form a coherent text. This also manifests as lower perplexity, indicating more fluent text.
>
> ---
>
> **W4, Q3:** *Practical issues in Real-world Applications*
>
> We acknowledge that the paper could benefit from a more detailed exploration of practical implementation challenges.
> We believe one of them is similar to that of graph foundation models, namely the issue of uniformity.
> Different graphs have varying text length limitations and text themes, and the LLMs’ understanding of them differs.
> In practice, it will be challenging to propose a unified text-level attack algorithm that does not require adjustments for each domain.
>
> Regarding scalability and complexity, please refer to the **author rebuttal**.
> In summary, scalability is not a major issue in current text-level GIAs, but it may need to be considered in future designs.
>
> ---
>
> **W6:** *Typos*
>
> Thank you for pointing out the typos. We will correct these in the revised version.
>
> ---
>
> **Q2:** *Application to Dynamic Graphs*
>
> We believe that attack and defense on dynamic graphs is a very interesting direction.
> When generating attacks on dynamic graphs, the historical interaction information of nodes should also be considered as a basis for generating injected nodes.
> At the same time, in dynamic graph scenarios, not all injected nodes are harmful.
> A more accurate characterization of this scenario will help us better align with real-world applications.
>
> ---
>
> Thank you once again for your valuable feedback and suggestions. We believe that incorporating these insights will significantly enhance our work’s quality and impact. We welcome any further questions or discussions.
>
> ---
>
> **Reference**
>
> [1] Chen, et al. Understanding and improving graph injection attack by promoting unnoticeability.
>
> [2] Chen, et al. Exploring the potential of large language models (LLMs) in learning on graphs.
>
> [3] Zheng, et al. Graph robustness benchmark: Benchmarking the adversarial robustness of graph machine learning.

---

> > ### Author Response · Authors · 2024-08-13
> >
> > Dear Reviewer,
> >
> > We sincerely appreciate the time and effort you have dedicated to reviewing our paper. Your feedback has been invaluable to us. We have provided detailed responses to the concerns and questions you raised during the rebuttal period. Your insights are crucial to enhancing the quality of our work.
> >
> > With the discussion period ending within one day, we hope we have thoroughly addressed all your concerns. Should you have any further questions or suggestions, we would be delighted to continue our discussion.
> >
> > Thank you for your support and guidance.
> >
> > Best regards,
> >
> > Submission15853 authors

---

### Official Review · Reviewer_YfqF · 2024-07-23

**Soundness:** 2
**Presentation:** 2
**Contribution:** 2
**Rating:** 4
**Confidence:** 4

**Summary:**

This paper explores the vulnerability of GNNs to Graph injection Attacks (GIAs), which involve injecting malicious nodes into a graph. This paper explores GIAs at the text level, presenting three types of GIA designs that inject textual content into the graphs. The significance of text interpretability in attack effectiveness and the defense against text-level GIAs are also explored.

**Strengths:**

1. The idea is straightforward and easy to follow.
2. The study of text-level graph injection attacks is interesting and practical.

**Weaknesses:**

1. In line 126–131, the paper claims that traditional methods focus mainly on the embedding level. However, some existing works (e.g., [1]) focus on the raw feature level. The authors fail to discuss why these works cannot be applied to this task.
2. Limited technical contribution. In this paper, three kinds of text-level GIA are proposed. However, I think all of them are simply combining both existing GIA techniques and language models, which are kind of incremental contributions.
3. I think the proposed text-level GIA is similar to the traditional GIA, especially when the nodes' texts are encoded into embeddings by LMs. Therefore, I think some existing defense methods against GIA can be also applied to this task. The authors need to discuss that.
4. The experiments only focus on small-scale datasets. More large-scale datasets need to be included.
5. Only vanilla GCN is used in the experiments, which may induce to biases to the conclusions. More advanced GNNs need to be included.



[1] Understanding and Improving Graph Injection Attack by Promoting Unnoticeability. ICLR 2022.

**Questions:**

Refer to weaknesses.

**Limitations:**

A section to discuss the limitations is missing.

---

> ### Author Rebuttal · Authors · 2024-08-06
>
> Thank you for your thoughtful feedback on our paper. We have addressed your concerns and provided clarifications below.
>
> ---
>
> **W1:** *Why previous works cannot be applied to this task.*
>
> We would like to clarify that by “Raw Feature,” we are referring to raw text, which represents the unembedded original features of TAGs.
> Previous GIAs, including the work referenced in [1], require the use of text embeddings during the generation process and are limited to injecting embeddings rather than raw text.
> Consequently, these methods cannot be directly applied to our task.
>
> To apply the method from [1] to this task, the embedding-text step must be completed, which is achieved by ITGIA in our paper.
> It advances embedding-level GIAs by transforming the embeddings generated in [1] back into raw text.
> However, we discovered several practical issues with the generated content, such as poor interpretability, loss of accuracy, and transferability challenges.
> These issues have not been addressed in previous embedding-level studies of GIAs and are found to be a significant challenge first identified by us.
>
> ---
>
> **W2:** *Limited technical contribution.*
>
> Firstly, the three proposed text-level GIAs require the introduction of previously unconsidered designs.
> While ITGIA builds on traditional GIAs, we additionally introduce normalization to align with the structure of normal embedding and discover the importance of interpretable regions.
> In VTGIA and WTGIA, we first introduce LLMs to Graph Adversarial Attacks, especially GIAs, ensuring interpretability during poisoning content generation.
> In VTGIA, we explore three different strategies for direct generation.
> In WTGIA, we transform FGSM into a sparsity-constrained version suited for raw text scenarios, unlike traditional FGSM, which samples features based on average sparsity.
> Additionally, WTGIA constructs the “Words to Text” task and uses masking to enhance attack performance.
>
> Second, our research pioneers the exploration of text-level GIAs, aiming to fill the gaps left by previous studies.
> We identify incomplete evaluations in earlier GIAs and propose three new lines for development, highlighting the inherent challenges in this domain.
> As an initial attempt, it is essential to explore techniques that appear to be straightforward.
> Our work focuses on understanding the landscape of text-level GIAs and identifying critical issues that need future attention, which sets the stage for more sophisticated advancements in subsequent research.
>
> Thus, the perceived technical limitations should not detract from the significance of our work.
>
> ---
>
> **W3:** *Limited Defense GNNs.*
>
> In the original paper, we discussed the EGNNGuard, which demonstrated significant defensive capabilities as described in [1].
> When incorporating HAO or increasing sparsity, attackers can achieve effective improvements at the embedding level.
> In the **author rebuttal**, we have added the performance of text-level GIAs against other defense models, including the Layernorm trick (LN) mentioned in [1] and common methods like GAT and GraphSAGE.
> Interestingly, LN’s effectiveness on text-level GIAs is not as strong as previously reported for embedding-level attacks.
> This is because embedding-level GIAs often exhibit abnormal norms, which LN exploits for defense. However, text-level GIAs derive embeddings from real text, avoiding structural anomalies and bypassing LN’s defenses.
>
> The above results indicate that re-evaluating defenses against GIAs at the text level is a promising research direction.
> We acknowledge that adding more defense models could make the evaluations more comprehensive.
> However, considering that we have supplemented with classical backbones and methods proven effective in [1], as well as discussed new defense mechanisms at the text level, we believe this sufficiently supports the conclusions presented in our paper.
>
> ---
>
> **W4:** *Large-scale Datasets.*
>
> In the **author rebuttal**, we include results for the Reddit and ogbn-arxiv datasets, the former containing over 30,000 nodes and the latter containing over 160,000 nodes.
> Note that current text-level GIAs do not increase complexity concerning the graph structure.
> The text generation process operates in $O(N_{{inj}})$, proportional to the number of injected nodes, which is significantly smaller than the total number of nodes in the original graphs.
> Consequently, WTGIA can complete text generation even for ogbn-arxiv within 8 hours without parallelization optimization, demonstrating high efficiency.
>
> ---
>
> **Limitations**
>
> In Section 5, we discussed the new challenges faced by text-level GIAs.
> Regarding more limitations of the paper, we believe that the current framework can be extended to better integrate graph structure and text generation to improve performance.
> We will address more about limitations in future versions.
>
> ---
>
> **More about Experimental Completeness**
>
> Due to the complexity of presenting results for **all combinations of datasets, attacks, defenses, and hyperparameters** (which would require more than 100 tables for presentation by estimation), we did not include all results in the original paper.
> We controlled the number of datasets and defense models selected to better present our findings.
> As **we did not emphasize the advantages of text-level GIA in terms of technical performance and efficiency**, we believe it is reasonable to choose the most commonly used datasets and victim models of traditional GIAs for evaluation.
>
> In the **author rebuttal**, we also included the running time, more embedding models, and a dataset from a new domain, Reddit.
> We will consider your suggestions and incorporate newly added experiments.
> If you have any further questions, we welcome more discussion.
>
> ---
>
> **Reference**
>
> [1] Chen, et al. Understanding and improving graph injection attack by promoting unnoticeability. ICLR 2022.

---

> > ### Author Response · Authors · 2024-08-13
> >
> > Dear Reviewer,
> >
> > We sincerely appreciate the time and effort you have dedicated to reviewing our paper. Your feedback has been invaluable to us. We have provided detailed responses to the concerns and questions you raised during the rebuttal period. Your insights are crucial to enhancing the quality of our work.
> >
> > With the discussion period ending within one day, we hope we have thoroughly addressed all your concerns. Should you have any further questions or suggestions, we would be delighted to continue our discussion.
> >
> > Thank you for your support and guidance.
> >
> > Best regards,
> >
> > Submission15853 authors

---

> > > ### Comment · Reviewer_YfqF · 2024-08-13
> > >
> > > Thanks for your responses. I read all the other reviewers' comments, I have adjusted my scores and will make final decisions after discussing with other reviewers and Acs.

---

> > > > ### Author Response · Authors · 2024-08-13
> > > >
> > > > Thank you for your valuable feedback and insights. We appreciate your time and effort in reviewing our work. Please feel free to reach out if you have any further questions or need additional information.

---

### Author Rebuttal · Authors · 2024-08-06

**Summary of Rebuttal**

Thanks to the reviewers for their diligent efforts and thorough evaluation.
We are glad to receive such positive feedback.
Notably, all reviewers acknowledged that conducting text-GIA is novel and meaningful.
Reviewers nfHh and Da4q recognized our method's technical contributions, while Reviewer ZG7Z also appreciated our method's contribution to ensuring the semantic integrity of the injection text.

The reviewers' main concerns focused on the comprehensiveness of our evaluation.
In response, during the rebuttal period, we provided additional analyses on defense methods, efficiency, results from more embedding methods, and experiments on additional datasets.
Specifically, In the author rebuttal and the uploaded PDF file, we have added the following content:

---

**Experiments Against More Defense Models**

In **Tables 1 to 6 in the pdf**, we included classic methods like GAT and GraphSAGE, as well as the EGNNGuard and Layernorm (LN) methods, which have been proven highly effective in [1].

We discovered interesting new results, such as LN’s effectiveness on text-level GIAs not being as strong as previously reported for embedding-level attacks, especially for WTGIA.
This is because embedding-level GIAs often exhibit abnormal norms, which LN exploits for defense.
However, text-level GIAs derive embeddings from real text, avoiding structural anomalies and bypassing LN’s defenses. This suggests that some traditional defense methods that were particularly effective may be limited to the embedding level.
The EGNNGuard method is generally effective and performs well overall.
As mentioned in the main paper, attackers can use techniques like HAO to bypass EGNNGuard, but this often involves trade-offs.

While including more baselines could offer a more comprehensive assessment, it entails a substantial workload in the domain of Graph Adversarial Attacks.
Based on our estimates, presenting results for **all combinations of datasets, attacks (both embedding-level and text-level), defenses, and hyperparameters would require over 100 tables**.
Therefore, we have chosen to focus on the most representative examples to provide a clearer and more concise analysis.

---

**Running Time and Complexity**

In **Tables 7 and 8 in the pdf**, we included the running time of the largest dataset, PubMed, from the original paper.
We also included results on ogbn-arxiv, which contains more than 160,000 nodes, to verify the scalability of our method.

In terms of graph structure, the text-level GIA proposed in the paper does not introduce additional complexity regarding graph structure.
Based on the results in [1], even the most complicated variant, MetaGIA, has a time complexity bounded by
$O(|V_T| N_{inj}^2 \log (|V_T| N_{inj}))$ and the space complexity is bounded by $O(|V_T| N_{inj})$

Considering that the number of injected nodes and edges is usually small compared to the original graph, current Text-GIAs are intrinsically scalable.
For example, on ogbn-arxiv, as long as the most complex MetaGIA is not used as the backbone, other experiments can be effectively completed.

In terms of text generation, our methods are linear to the number of injected nodes and can be executed efficiently.
For all methods that rely on LLM, we can further improve efficiency using techniques like LLM's inference acceleration, parallel generation, and reducing the number of correction rounds.

Therefore, scalability is not our bottleneck at present.

---

**More Embeddings**

In **Tables 9 and 10 in the pdf**, we included SentenceBert all-MiniLM-L6-v2 [2] as a new embedding backbone.
We find that WTGIA shows some transferability between different embeddings, but the results are still unsatisfactory. Although GTR and SBERT are both PLM-based embeddings, the adversarial text by ITGIA performs even worse than WTGIA when transferred to SBERT.
We believe the transferability of text-level GIAs remains a significant challenge.

---

**More Datasets**

In **Tables 11 and 12 of the PDF**, we included the ogbn-arxiv dataset [3], containing over 160,000 nodes, and a social network dataset, Reddit, with more than 30,000 nodes [4], to evaluate the scalability and generalization of text-level GIAs.
We injected 1,500 nodes into ogbn-arxiv, following the budget specified in [1], and proportionally injected 500 nodes into Reddit.
Due to space constraints, we use WTGIA with average sparsity budgets, as it has the longest runtime and best performance.

On ogbn-arxiv, WTGIA maintains use rates above 90% and shows good attack performance when transferred to GTR.
On Reddit, though effective on BoW embeddings, the transferability issue is more severe, indicating domain variability in graph attacks.
With more datasets from various domains becoming available in the future, we look forward to exploring the differences in graph attacks and defenses across different domains.

---

**Update Plan**

We addressed other weaknesses and questions in each separate rebuttal.
We will incorporate the above content in the updated version of our paper.
In addition, we will improve explanations of interpretability and the embedding-text-embedding process, correct typos, and add necessary citations based on the reviewers’ suggestions.

We sincerely thank the reviewers for their valuable feedback and constructive comments, which have significantly contributed to improving our work.
If there are any further questions or if you would like additional clarification, we welcome further discussion and engagement.

---

**Reference**

[1] Chen, et al. Understanding and improving graph injection attack by promoting unnoticeability.

[2] Reimers, Nils, and Iryna Gurevych. Sentence-BERT: Sentence embeddings using Siamese BERT-networks.

[3] Hu, et al. Open graph benchmark: Datasets for machine learning on graphs.

[4] Huang et al. Can GNN be a good adapter for LLMs?

---

### Decision · Program_Chairs · 2024-09-25

**Decision:**

Accept (poster)

**Comment:**

It seems to me that all major concerns of the reviewers have been addressed during the discussion period and that the paper merits acceptance at the NeurIPS conference.

The reviews and the rebuttal have given rise to several interesting points and results that I encourage the authors to include in the camera-ready version. Especially, the multitude of additional experimental results that the authors added in the general author rebuttal would be a valuable addition to the paper, since the evaluation on only three datasets, seems to be insufficient to me too. I furthermore want to encourage the authors to add even more defense baseline models to their camera-ready version (as several reviewers also requested). While the new LayerNorm and EGNNGuard results provided in the rebuttal are interesting, I feel that they still do not do justice to the vast literature that exists on defense methodology to adversarial attacks on graph structure and node features. The impact of this paper could be improved if the method was benchmarked against a broader range of defense methodologies.